# NODE IDENTIFIERS: COMPACT, DISCRETE REPRESENTATIONS FOR EFFICIENT GRAPH LEARNING

**Yuankai Luo**[1,2]     **Hongkang Li**[3*]     **Qijiong Liu**[2*]     **Lei Shi**[1†]     **Xiao-Ming Wu**[2†]

[1]Beihang University   [2]The Hong Kong Polytechnic University   [3]Rensselaer Polytechnic Institute

## ABSTRACT

We present a novel end-to-end framework that generates highly compact (typically 6-15 dimensions), discrete (int4 type), and interpretable node representations—termed node identifiers (node IDs)—to tackle inference challenges on large-scale graphs. By employing vector quantization, we compress continuous node embeddings from multiple layers of a Graph Neural Network (GNN) into discrete codes, applicable under both self-supervised and supervised learning paradigms. These node IDs capture high-level abstractions of graph data and offer interpretability that traditional GNN embeddings lack. Extensive experiments on 34 datasets, encompassing node classification, graph classification, link prediction, and attributed graph clustering tasks, demonstrate that the generated node IDs significantly enhance speed and memory efficiency while achieving competitive performance compared to current state-of-the-art methods.

## 1 INTRODUCTION

Machine learning on graphs involves leveraging graph topology and node/edge attributes to perform various tasks, including node and graph classification (Hu et al., 2020; Luo et al., 2023b), link prediction (Lü & Zhou, 2011; Zhang & Chen, 2018), community detection (Fortunato, 2010; Fortunato & Hric, 2016), and recommendation (Konstas et al., 2009). Various methods have been developed to address these challenges, including random walk based-models (Perozzi et al., 2014; Grover & Leskovec, 2016), spectral methods (Bruna et al., 2013; Defferrard et al., 2016), and graph neural networks (GNNs) (Hamilton et al., 2017; Kipf & Welling, 2017; Veličković et al., 2018a). GNNs employ a message-passing mechanism (Gilmer et al., 2017) to iteratively aggregate information from a node's neighbors. This process enables GNNs to learn node representations by effectively integrating graph topology and node attributes, leading to impressive results across various tasks.

Despite the advancements in GNNs, their application to large-scale scenarios requiring low latency and fast inference remains challenging (Zhang et al., 2020; Jia et al., 2020; Yang et al., 2024). The inherent bottleneck is the message-passing mechanism, which necessitates loading the entire graph—potentially comprising billions of edges—during inference for target nodes, which is computationally demanding and time-consuming. To address this challenge, recent studies (Zhang et al., 2021b; Tian et al., 2022; Yang et al., 2024) have explored knowledge distillation techniques to distill a small MLP model that captures essential information from a pre-trained GNN, making it suitable for latency-critical applications (Tian et al., 2023). However, these GNN-to-MLP methods require supervised training using class labels and lack the ability to generate effective node representations.

An alternative approach to facilitate inference on large-scale graphs involves learning effective, low-dimensional node embeddings that can be directly utilized for downstream prediction tasks. This method has been extensively explored in early works on network/graph embedding, such as DeepWalk (Perozzi et al., 2014) and node2vec (Grover & Leskovec, 2016), as well as in recent efforts leveraging GNNs to learn embeddings for graph tokens (Liu et al., 2024d;a; Qu et al., 2024; Xia et al., 2023), including nodes, edges, and (sub)graphs. To balance representation effectiveness and computational efficiency, embeddings generated by GNNs typically have dimensions of 128 or 256. However, this

---

*Contributed equally as the second authors.

†Corresponding authors. Correspondence to: luoyk@buaa.edu.cn, xiao-ming.wu@polyu.edu.hk.

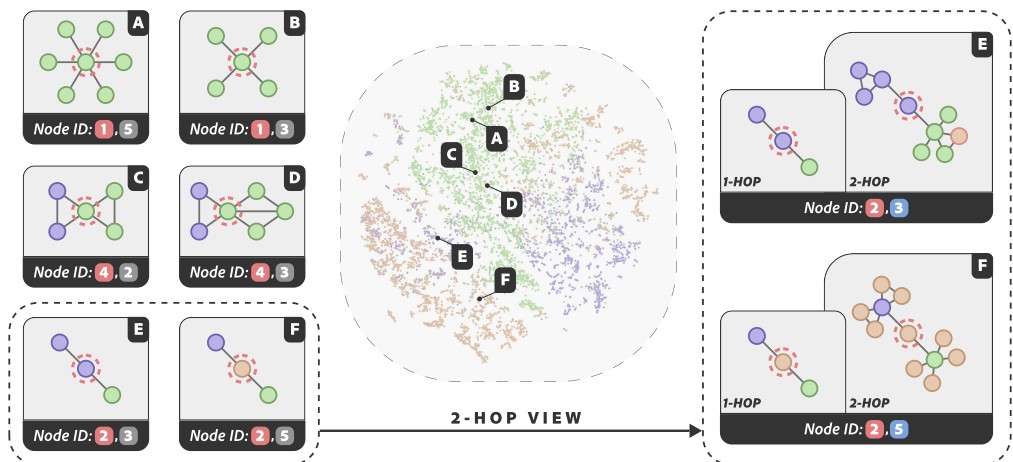

Figure 1: Illustration of 2-dimensional node IDs generated by our NID framework using a two-layer GCN. The first ID code is derived from the first-layer node embeddings, and the second ID code from the second-layer embeddings. *Center:* t-SNE visualization of node embeddings in the PubMed Dataset, with colors indicating different class labels. *Left:* Six nodes are shown with their IDs and 1-hop neighborhoods (topology and label distribution). Nodes sharing the same first ID code have similar 1-hop neighborhoods, though not necessarily the same class label. *Right:* Nodes E and F are further analyzed with their 2-hop neighborhoods, where differences are reflected in their distinct second ID code (blue) and class label.

relatively high dimensionality poses challenges in terms of storage and computational efficiency for large-scale applications. Furthermore, the real-valued embeddings often lack interpretability.

In this work, we tackle this challenge by learning highly compact (typically 6-15 dimensions), discrete (int4 type), and interpretable node representations, termed node identifiers (node IDs). We introduce a novel end-to-end framework called NID, which employs vector quantization (Gray, 1984) to compress the continuous node embeddings generated at each layer of a GNN into discrete codes, where the GNN and codebooks are trained jointly. This approach effectively captures the multi-order neighborhood structures within the graph. Fig. 1 illustrates examples of two-dimensional node IDs generated by a two-layer GNN, demonstrating their ability to capture multi-level structural patterns, as well as their interpretability. We summarize the significance of our work as follows:

- We empirically and theoretically demonstrate the feasibility of learning highly compact, discrete codes (node IDs) as effective node representations for efficient graph learning, without relying on knowledge distillation. Our extensive evaluation across 34 diverse datasets and tasks—including node and graph classification, link prediction, and attributed graph clustering—shows that these node IDs achieve performance competitive with state-of-the-art methods while significantly enhancing speed and memory efficiency. We also offer a theoretical justification for our approach.

- Our proposed NID framework can be integrated with state-of-the-art unsupervised and supervised GNN methods to enhance performance. Experiments demonstrate that the generated node IDs not only retain essential information from GNN embeddings but also uncover hidden patterns in some cases, leading to significantly improved performance. This warrants deeper investigation.

- Our findings indicate significant redundancy in GNN embeddings. The generated compact, discrete node IDs provide a high-level abstraction of graph data, offering interpretability that GNN embeddings lack. This may facilitate graph tokenization and applications involving LLMs.

## 2 PRELIMINARIES

We define a graph as a tuple $\mathcal{G} = (\mathcal{V}, \mathcal{E}, \boldsymbol{X})$, where $\mathcal{V}$ is the set of nodes, $\mathcal{E} \subseteq \mathcal{V} \times \mathcal{V}$ is the set of edges, and $\boldsymbol{X} \in \mathbb{R}^{|\mathcal{V}| \times d}$ is the node feature matrix, with $|\mathcal{V}|$ representing the number of nodes and $d$ the dimension of the node features. Let $\boldsymbol{A} \in \mathbb{R}^{|\mathcal{V}| \times |\mathcal{V}|}$ denote the adjacency matrix of $\mathcal{G}$.

**Message Passing Neural Networks (MPNNs)** have become the dominant approach for learning graph representations. A typical example is graph convolutional networks (GCNs) (Kipf & Welling, 2017). Gilmer et al. (2017) reformulated early GNNs into a framework of message passing GNNs, which computes representations $\boldsymbol{h}_v^l$ for any node $v$ in each layer $l$ as:

$$\boldsymbol{h}_v^l = \text{UPDATE}^l\left(\boldsymbol{h}_v^{l-1}, \text{AGG}^l\left(\left\{\boldsymbol{h}_u^{l-1} \mid u \in \mathcal{N}(v)\right\}\right)\right), \tag{1}$$

where $\mathcal{N}(v)$ denotes the neighborhood of $v$, $\text{AGG}^l$ is the message function, and $\text{UPDATE}^l$ is the update function. The initial node representation $\boldsymbol{h}_v^0$ is the node feature vector $\boldsymbol{x}_v \in \mathbb{R}^d$. The message function aggregates information from the neighbors of $v$ to update its representation. The output of the last layer, i.e., $\text{MPNN}(v, \boldsymbol{A}, \boldsymbol{X}) = \boldsymbol{h}_v^L$, is the representation of $v$ produced by the MPNN.

Prediction tasks on graphs involve node-level, edge-level, and graph-level tasks. Each type of tasks requires a tailored graph readout function, R, which aggregates the output node representations, $\boldsymbol{h}_v^L$, from the last layer $L$, to compute the final prediction result:

$$\boldsymbol{h}_{\text{readout}} = \text{R}\left(\left\{\boldsymbol{h}_v^L, v \in \mathcal{V}\right\}\right). \tag{2}$$

Specifically, for *node-level* tasks, which involve classifying individual nodes, R is simply an identity mapping. For *edge-level* tasks, which focus on analyzing the relationship between any node pair $(u, v)$, R is typically modeled as the Hadamard product of the node representations (Kipf & Welling, 2016), i.e., $\boldsymbol{h}_{\text{readout}} = \boldsymbol{h}_v^L \odot \boldsymbol{h}_u^L$. For *graph-level* tasks that aim to make predictions about the entire graph, R often functions as a global mean pooling operation, expressed as $\boldsymbol{h}_{\text{readout}} = \frac{1}{|\mathcal{V}|} \sum_{v \in \mathcal{V}} \boldsymbol{h}_v^L$.

**Vector Quantization (VQ)** (Gray, 1984) aims to represent a large set of vectors, $\boldsymbol{Z} = \{\boldsymbol{z}_i\}_{i=1}^N$, with a small set of prototype (code) vectors of a codebook $\boldsymbol{C} = \{\boldsymbol{e}_k\}_{k=1}^K$, where $N \gg K$. The codebook is often created using algorithms such as $k$-means clustering via optimizing the following objective:

$$\textbf{VQ:} \quad \min_{\boldsymbol{C}} \sum_{i=1}^N \min_{k=1}^K ||\boldsymbol{z}_i - \boldsymbol{e}_k||_2^2. \tag{3}$$

Once the codebook is learned, each vector $\boldsymbol{z}_i$ can be approximated by its closet prototype vector $\boldsymbol{e}_t$, where $t = \arg\min_k ||\boldsymbol{z}_i - \boldsymbol{e}_k||_2^2$ is the index of the prototype vector. **Residual Vector Quantization (RVQ)** (Juang & Gray, 1982; Martinez et al., 2014) extends the basic VQ by performing the quantization process $M$ times. Initially, a standard VQ is performed. Then, the *residual vector* is calculated:

$$\boldsymbol{r}_i = \boldsymbol{z}_i - \boldsymbol{e}_t, \tag{4}$$

which represents the quantization error from the initial quantization. Then, the residual vectors $\boldsymbol{r}_i$ are quantized using a second codebook. This process can be performed repeatedly, with each stage quantizing the residual error from the previous stage, ultimately producing $M$ tiers of codes.

## 3  OUR PROPOSED NODE ID (NID) FRAMEWORK

Our proposed NID framework consists of two stages:

1. *Generating compact, discrete node IDs.* Nodes are encoded using multi-layer MPNNs to capture multi-order neighborhood structures. At each layer, the node embedding is quantized into a tuple of structural codewords. The tuples are then combined to form what we refer to as the node ID.

2. *Utilizing the generated node IDs as node representations in various downstream tasks.* We directly use the node IDs for unsupervised tasks such as node clustering. We train simple MLPs with the node IDs for supervised tasks including node classification, link prediction, and graph classification.

### 3.1  GENERATION OF NODE IDs

Fig. 2 illustrates the diverse clustering patterns of node representations produced by an MPNN at different layers $l$. This diversity arises from the cumulative smoothing effect caused by successive applications of graph convolution at each layer (Li et al., 2018; 2019). To generate structure-aware node IDs, we employ an $L$-layer MPNN to capture multi-order neighborhood structures. At each layer, we use vector quantization to encode the node embeddings produced by the MPNN into

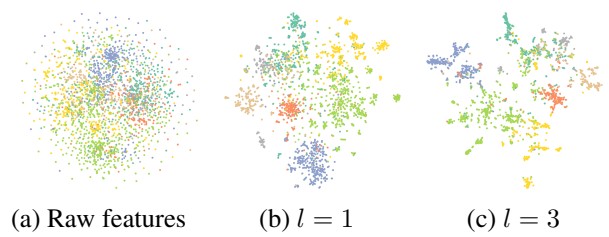

(a) Raw features      (b) $l = 1$      (c) $l = 3$

Figure 2: t-SNE visualization of the node representations of the Cora dataset generated by an MPNN at different layers $l$.

Figure 3: Overview of our proposed NID framework.

$M$ codewords (integer indices). For each node $v$, we define the node ID of $v$ as a tuple composed of $L \times M$ codewords, structured as follows:

$$\text{Node\_ID}(v) = (c_{11}, \cdots, c_{1M}, c_{21}, \cdots, c_{2M}, \cdots, c_{L1}, \cdots, c_{LM}), \tag{5}$$

where $c_{lm}$ represents the $m$-th codeword at the $l$-th layer. Both $M$ and $L$ can be very small. For the node IDs in Fig. 1, $M = 1$ and $L = 2$. In our experiments, we typically set $M = 3$ and $L \in [2, 5]$.

**Learning Node IDs.** As illustrated in Fig. 3, at each layer $l$ ($1 \le l \le L$) of the MPNN, we employ RVQ to quantize the node embeddings and produce $M$ tiers of codewords for each node $v$ (see App. E.1 for a detailed explanation). Each codeword $c_{lm}$ ($1 \le m \le M$) is generated by a *distinct* codebook $\boldsymbol{C}_{lm} = \{\boldsymbol{e}_k^{lm}\}_{k=1}^K$, where $K$ is the size of the codebook. Hence, there are a total of $L \times M$ codebooks, indexed by $lm$. Let $\boldsymbol{r}_{lm}$ denote the vector to be quantized. Note that $\boldsymbol{r}_{l1}$ is the node embedding $\boldsymbol{h}_v^l$ produced by the MPNN. When $m > 1$, $\boldsymbol{r}_{lm}$ represents the residual vector. Then, $\boldsymbol{r}_{lm}$ is approximated by its nearest code vector from the corresponding codebook $\boldsymbol{C}_{lm}$:

$$c_{lm} = \arg\min_k ||\boldsymbol{r}_{lm} - \boldsymbol{e}_k^{lm}||, \tag{6}$$

producing the codeword $c_{lm}$, which is the index of the nearest code vector.

We introduce a simple, generic framework for learning node IDs (codewords $c_{lm}$) by jointly training the MPNN and the codebooks with the following loss function:

$$\mathcal{L}_{\text{NID}} = \mathcal{L}_{\mathcal{G}} + \mathcal{L}_{\text{VQ}}, \tag{7}$$

where $\mathcal{L}_{\mathcal{G}}$ is a (self-)supervised graph learning objective, and $\mathcal{L}_{\text{VQ}}$ is a vector quantization loss. $\mathcal{L}_{\mathcal{G}}$ aims to train the MPNN to produce effective node embeddings, while $\mathcal{L}_{\text{VQ}}$ ensures the codebook vectors align well with the node embeddings. For a single node $v$, $\mathcal{L}_{\text{VQ}}$ is defined as

$$\mathcal{L}_{\text{VQ}} = \sum_{l=1}^{L} \sum_{m=1}^{M} ||\text{sg}(\boldsymbol{r}_{lm}) - \boldsymbol{e}_{c_{lm}}^{lm}|| + \beta ||\boldsymbol{r}_{lm} - \text{sg}(\boldsymbol{e}_{c_{lm}}^{lm})||, \tag{8}$$

where sg denotes the stop gradient operation, and $\beta$ is a weight parameter. The first term in Eq. (8) is the *codebook loss* (Van Den Oord et al., 2017), which only affects the codebook and brings the selected code vector close to the node embedding. The second term is the *commitment loss* (Van Den Oord et al., 2017), which only influences the node embedding and ensures the proximity of the node embedding to the selected code vector. In practice, we can use exponential moving averages (Razavi et al., 2019) as a substitute for the *codebook loss*.

**Self-supervised Learning.** The graph learning objective $\mathcal{L}_{\mathcal{G}}$ can be a self-supervised learning task, such as graph reconstruction (i.e., reconstructing the node features or adjacency matrix) or contrastive learning (Liu et al., 2021). In this paper, we examine two representative models: GraphMAE (Hou et al., 2022) and GraphCL (You et al., 2020). We discuss GraphMAE here and address GraphCL in the App. A due to space limitations. Specifically, GraphMAE involves sampling a subset of nodes $\tilde{\mathcal{V}} \subset \mathcal{V}$, masking the node features as $\tilde{\boldsymbol{X}}$, encoding the masked node features using an MPNN, and subsequently reconstructing the masked features with a decoder. The reconstruction loss is based on the scaled cosine error, expressed as:

$$\mathcal{L}_{\mathcal{G}} = \mathcal{L}_{\text{MAE}} = \frac{1}{|\tilde{\mathcal{V}}|} \sum_{v \in \tilde{\mathcal{V}}} \left(1 - \frac{\boldsymbol{x}_v^T \boldsymbol{z}_v}{||\boldsymbol{x}_v|| \cdot ||\boldsymbol{z}_v||} \cdot \gamma\right),$$

where $\tilde{\mathcal{V}}$ is the set of masked nodes, $\boldsymbol{z}_v = f_D(\tilde{\boldsymbol{h}}_v^L)$ is the reconstructed node features by a decoder $f_D$, $\tilde{\boldsymbol{h}}_v^L = \text{MPNN}(v, \boldsymbol{A}, \tilde{\boldsymbol{X}})$, and $\gamma \ge 1$ is a scaling factor. Let $\tilde{\boldsymbol{r}}_{l1} := \tilde{\boldsymbol{h}}_v^l$ denote the node embedding generated by the $l$-th layer of the MPNN with the masked features. The overall training loss is

$$\mathcal{L}_{\text{NID}} = \mathcal{L}_{\text{MAE}} + \sum_{v \in \tilde{\mathcal{V}}} \sum_{l=1}^{L} \sum_{m=1}^{M} \|\text{sg}(\tilde{\boldsymbol{r}}_{lm}) - \boldsymbol{e}_{c_{lm}}\| + \beta \|\tilde{\boldsymbol{r}}_{lm} - \text{sg}(\boldsymbol{e}_{c_{lm}})\|. \tag{9}$$

**Supervised Learning.** The graph learning objective $\mathcal{L}_{\mathcal{G}}$ can also be a supervised learning task, such as node classification, link prediction, or graph classification. For classification tasks, $\mathcal{L}_{\mathcal{G}}$ can be the cross-entropy loss between the target label $y$ and the prediction $\boldsymbol{h}_{\text{readout}}$ (see Eq. (2)):

$$\mathcal{L}_{\mathcal{G}} = \mathcal{L}_{\text{CE}}(y, \boldsymbol{h}_{\text{readout}}). \tag{10}$$

**Remark.** Our NID framework differs from VQ-VAE (Van Den Oord et al., 2017) and similar approaches (Lee et al., 2022; Yang et al., 2024) in codebook learning. Unlike these methods, our training objective $\mathcal{L}_{\text{NID}}$ does not involve using the code vectors ($\boldsymbol{e}_k$) for a reconstruction task. Instead, we guide the codebook learning process solely via graph learning tasks ($\mathcal{L}_{\mathcal{G}}$). This is because our experiments show that omitting the reconstruction loss has a negligible impact on performance (see Appendix D.5 for details). Moreover, our NID framework is compatible with any MPNN model. In experiments, we use popular models like GCN (Kipf & Welling, 2017), GAT (Veličković et al., 2018a), SAGE (Hamilton et al., 2017) and GIN (Xu et al., 2018).

## 3.2 APPLICATIONS OF NODE IDS FOR GRAPH LEARNING

The generated node IDs can be considered as highly compact node representations and used directly for various downstream graph learning tasks, as outlined below.

**Node-level tasks** include *node classification* and *node clustering*. For node classification, each node $v$ in the graph is associated with a label $y_v$, representing its category. We can directly utilize the node IDs of the labeled nodes to train an MLP network for classification. The prediction is formulated as

$$\hat{y}_v = \text{MLP}(\text{Node\_ID}(v)). \tag{11}$$

For node clustering, one can directly apply vector-based clustering algorithms such as $k$-means (MacQueen et al., 1967) to the node IDs to obtain clustering results.

**Edge-level tasks** typically involve *link prediction*. The aim is to predict whether an edge should exist between any node pair $(u, v)$. The prediction can be made by

$$\hat{y}_{(u,v)} = \text{MLP}(\text{Node\_ID}(u) \odot \text{Node\_ID}(v)), \tag{12}$$

where $\odot$ is the Hadamard product.

**Graph-level tasks** include *graph classification* and *graph regression*. These tasks involve predicting a categorical label or numerical value for the entire graph $\mathcal{G}$. The prediction can be formulated as

$$\hat{y}_{\mathcal{G}} = \text{MLP}\left(\frac{1}{|\mathcal{V}|} \sum_{v \in \mathcal{V}} \text{Node\_ID}(v)\right), \tag{13}$$

where a global mean pooling function is applied on all the node IDs to generate a representation for the graph $\mathcal{G}$, which is then input into an MLP for prediction. Note that the selection of the readout function, such as mean pooling, is considered a hyper-parameter.

**Remark.** Due to the high compactness of the node IDs, which usually consists of multiple codewords (int4 type in our experiments), the inference process of the aforementioned graph learning tasks can be *greatly accelerated*. Furthermore, as illustrated in Fig. 1, the node IDs represent a high-level abstraction of structure-aware information in a graph, enabling them to achieve competitive performance across various tasks, as evidenced in our evaluation.

## 3.3 THEORETICAL ANALYSIS

We provide a theoretical analysis to verify the validity of the proposed method with a simplified model. We prove the optimized codebook by VQ can distinguish nodes based on a widely used data formulation, which leads to a desired classification performance by training a linear layer.

**Theoretical formulation.** Consider a node-level $P$-classification problem on a graph $\mathcal{G} = \{\mathcal{V}, \mathcal{E}, \boldsymbol{X}\}$ with $\boldsymbol{A} \in \mathbb{R}^{|\mathcal{V}| \times |\mathcal{V}|}$ as the adjacency matrix and $\boldsymbol{Y}$ as the labels. We use a one-layer GCN to generate

Table 1: Node classification results in supervised representation learning over homophilic and heterophilic graphs (%). The baseline results are primarily taken from Polynormer (Deng et al., 2024).

| Transductive | Cora | CiteSeer | PubMed | Computer | Photo | CS | Physics | WikiCS | Squirrel | Chameleon | Ratings | Questions |
|---|---|---|---|---|---|---|---|---|---|---|---|---|
| # nodes | 2,708 | 3,327 | 19,717 | 13,752 | 7,650 | 18,333 | 34,493 | 11,701 | 2223 | 890 | 24,492 | 48,921 |
| # edges | 5,278 | 4,732 | 44,324 | 245,861 | 119,081 | 81,894 | 247,962 | 216,123 | 46,998 | 8,854 | 32,927 | 153,540 |
| Metric | Accuracy↑ | Accuracy↑ | Accuracy↑ | Accuracy↑ | Accuracy↑ | Accuracy↑ | Accuracy↑ | Accuracy↑ | Accuracy↑ | Accuracy↑ | Accuracy↑ | ROC-AUC↑ |
| GPRGNN | 87.95 ±1.18 | 77.13 ±1.67 | 87.54 ±0.38 | 89.32 ±0.29 | 94.49 ±0.14 | 95.13 ±0.09 | 96.85 ±0.08 | 78.12 ±0.23 | 38.95 ±1.99 | 39.93 ±3.30 | 44.88 ±0.34 | 55.48 ±0.91 |
| APPNP | 87.87 ±0.82 | 76.53 ±1.16 | 88.43 ±0.15 | 90.18 ±0.17 | 94.32 ±0.14 | 94.49 ±0.07 | 96.54 ±0.07 | 78.87 ±0.11 | 36.88 ±1.27 | 41.62 ±3.13 | 52.74 ±0.73 | 77.82 ±1.31 |
| SGFormer | 87.83 ±0.92 | 77.24 ±0.74 | 89.31 ±0.54 | 92.42 ±0.66 | 95.58 ±0.36 | 95.71 ±0.24 | 96.75 ±0.26 | 80.05 ±0.46 | 42.65 ±2.41 | 45.21 ±3.72 | 54.14 ±0.62 | 73.81 ±0.59 |
| Polynormer | 88.11 ±1.08 | 76.77 ±1.01 | 87.34 ±0.43 | 93.18 ±0.18 | 96.11 ±0.23 | 95.51 ±0.29 | 97.22 ±0.06 | 79.53 ±0.83 | 40.87 ±1.96 | 41.82 ±3.45 | 54.46 ±0.40 | 78.92 ±0.89 |
| Graph-MLP | 87.06 ±1.38 | 76.43 ±1.44 | 88.93 ±0.63 | 90.78 ±0.41 | 95.43 ±0.76 | 94.68 ±0.28 | 95.45 ±0.24 | 75.35 ±0.55 | - | - | - | - |
| VQGraph | 86.11 ±1.26 | 75.64 ±0.92 | 88.03 ±0.63 | 90.28 ±0.47 | 94.98 ±0.59 | 93.82 ±0.17 | 95.93 ±0.28 | 77.92 ±0.61 | - | - | - | - |
| GCN | 88.77 ±0.61 | 77.53 ±0.92 | 90.04 ±0.25 | 93.78 ±0.31 | 96.14 ±0.21 | 95.94 ±0.28 | 97.36 ±0.07 | 80.91 ±0.81 | 44.50 ±1.92 | 46.11 ±3.16 | 53.57 ±0.32 | 77.40 ±1.07 |
| **NID**$_{GCN}$ | 87.88 ±0.69 | 76.89 ±1.09 | 89.42 ±0.44 | 93.41 ±0.08 | 96.17 ±0.04 | 95.52 ±0.10 | 97.34 ±0.04 | 78.55 ±0.15 | **45.09** ±1.72 | **46.29** ±2.92 | 53.55 ±0.13 | 96.85 ±0.10 |
| GAT | 88.22 ±1.24 | 77.08 ±0.84 | 89.47 ±0.25 | 93.53 ±0.18 | 96.27 ±0.15 | 94.46 ±0.14 | 97.17 ±0.09 | 80.98 ±0.83 | 38.72 ±1.46 | 43.44 ±3.00 | 54.88 ±0.74 | 78.35 ±1.16 |
| **NID**$_{GAT}$ | 87.35 ±0.57 | 76.13 ±1.35 | 88.97 ±0.36 | 93.38 ±0.16 | **96.47** ±0.27 | 94.75 ±0.16 | 97.13 ±0.08 | 79.56 ±0.43 | 37.68 ±2.04 | 42.83 ±3.42 | **54.92** ±0.42 | **97.03** ±0.02 |

the node embedding $\boldsymbol{h}_v$ for $v \in \mathcal{V}$, i.e., $\boldsymbol{h}_v = \sigma\left(\frac{1}{|\mathcal{N}(u)|} \sum_{u \in \mathcal{N}(u)} \boldsymbol{W} \boldsymbol{x}_u\right)$, where $\boldsymbol{W} \in \mathbb{R}^{d' \times d}$ and we assume $\sigma(\cdot)$ as an identity function. Given the obtained natural number $\text{Node\_ID}(v) \in \mathbb{N}^r$ for node $v \in \mathcal{V}$, $r < K$, we map each entry to a $d_e$-dimensional embedding, where different digits correspond to orthogonal embeddings. Hence, $\text{Node\_ID}(v)$ is projected to a $rd_e$-dimensional vector, denoted by $\boldsymbol{z}_v$. We use nodes from $\mathcal{V}_R$ to train a linear head $\boldsymbol{V} \in \mathbb{R}^{rd_e \times P}$, i.e., $\min_{\boldsymbol{V}} \frac{1}{|\mathcal{V}_R|} \sum_{v \in \mathcal{V}_R} \ell(\boldsymbol{z}_v, y_v; \boldsymbol{V})$, where $\ell(\boldsymbol{z}_v, y_v; \boldsymbol{V}) = -\boldsymbol{y}_i^\top \log(\hat{\boldsymbol{p}}_v) := -\boldsymbol{y}_i^\top \log(\text{softmax}(\boldsymbol{z}_v^\top \boldsymbol{V}))$. Here $\boldsymbol{y}_i$ is the one-hot vector of $y_i \in \mathbb{N}$, where only the $y_i + 1$-th entry is 1 and others are 0. The classification error is defined by $\mathbb{1}[y_v \neq \arg\max_{i \in [P]} \hat{\boldsymbol{p}}_{v,i}]$ for $v \in \mathcal{V}$, where $\hat{\boldsymbol{p}}_{v,i}$ is the $i$-th entry of $\hat{\boldsymbol{p}}_v$.

We define a set of orthonormal vectors $\{\boldsymbol{\mu}_i\}_{i=1}^Q$. For a $P$-classification problem ($P < Q$), denote $\{\boldsymbol{\mu}_i\}_{i=1}^P$ as the set of discriminative patterns that directly determine the label, while $\{\boldsymbol{\mu}_i\}_{i=P+1}^Q$ as the set of non-discriminative patterns that are irrelevant to the label. Specifically, we assume that for any node $v \in \mathcal{V}$ with label $p \in [P]$, $\boldsymbol{x}_v = \boldsymbol{\mu}_p$ or at least one of its neighbor equals $\boldsymbol{\mu}_p$, and no neighbor of $\boldsymbol{x}_v$ or itself equals $\boldsymbol{\mu}_j$ for $j \in [P], j \neq p$. This formulation extends from binary node classification in (Zhang et al., 2023b; Li et al., 2024), which is verified on real-world datasets. We also assume rows of the optimized $\boldsymbol{W}$ by (7) are in directions of $\boldsymbol{\mu}_i, i \in [P]$ uniformly by theoretical findings in (Zhang et al., 2023b; Li et al., 2024). Then, we have the following theorem.

**Theorem 1.** *The optimizer $\boldsymbol{C}^*$ of VQ objective (7) satisfies that, for any $\boldsymbol{x}_u$ and $\boldsymbol{x}_v$, $u, v \in \mathcal{V}$ with different labels, $\text{Node\_ID}(u) \neq \text{Node\_ID}(v)$. Then, as long as $\mathcal{V}_R$ uniformly include node IDs from all the classes, by training the linear head $\boldsymbol{V}$ with sufficient gradient descent steps, we can achieve that the classification error $\mathbb{1}[y_v \neq \arg\max_{i \in [P]} \hat{\boldsymbol{p}}_{v,i}] = 0$ for any $v \in \mathcal{V}$.*

**Remark.** Theorem 1 illustrates that the optimized $\boldsymbol{C}^*$ of VQ objective (7) ensures that the obtained IDs from different classes are distinct. Then, we demonstrate that with node IDs in the training set, a linear head can be learned to achieve a zero classification error. The proof can be found in App. B.

## 4 EVALUATION

In this section, we demonstrate the versatility of our NID framework across various graph learning tasks. We detail its application in two distinct scenarios:

- **Supervised representation learning for node classification, link prediction and graph classification.** Here, we evaluate our NID against several SOTA models for graph representation learning, following learning protocols (Luo et al., 2024b; Rampášek et al., 2022; Wang et al., 2023c).
- **Unsupervised representation learning for attributed graph clustering, node classification, and graph classification.** In these unsupervised tasks, NID is benchmarked against well-known contrastive and generative SSL methods. We adhere strictly to the established experimental procedures as the standard settings (Bhowmick et al., 2024; Hou et al., 2022; You et al., 2020).

Detailed datasets, baselines, and hyperparameters are provided in App. C due to space constraints.

### 4.1 OVERALL PERFORMANCE

The learned node IDs, **typically comprising 6 to 15 int4 integers**, serve as effective node representations. They achieve competitive or superior performance across a wide range of tasks while significantly enhancing speed and memory efficiency. Additionally, our NID framework can be integrated with SOTA supervised or unsupervised GNN methods to enhance performance.

Table 2: Node classification results in supervised representation learning on large-scale graphs (%).

| Transductive | ogbn-proteins | ogbn-arxiv | ogbn-products | pokec |
|---|---|---|---|---|
| # nodes | 132,534 | 169,343 | 2,449,029 | 1,632,803 |
| # edges | 39,561,252 | 1,166,243 | 61,859,140 | 30,622,564 |
| Metric | ROC-AUC↑ | Accuracy↑ | Accuracy↑ | Accuracy↑ |
| GPRGNN | 75.68 $\pm$ 0.49 | 71.10 $\pm$ 0.12 | 79.76 $\pm$ 0.59 | 78.83 $\pm$ 0.05 |
| LINKX | 71.37 $\pm$ 0.58 | 66.18 $\pm$ 0.33 | 71.59 $\pm$ 0.71 | 82.04 $\pm$ 0.07 |
| GraphGPS | 76.83 $\pm$ 0.26 | 70.97 $\pm$ 0.41 | OOM | OOM |
| SGFormer | 79.53 $\pm$ 0.38 | 72.63 $\pm$ 0.13 | 74.16 $\pm$ 0.31 | 73.76 $\pm$ 0.24 |
| Polynormer | 75.97 $\pm$ 0.47 | 71.82 $\pm$ 0.23 | 82.97 $\pm$ 0.28 | 85.95 $\pm$ 0.07 |
| SAGE | 79.43 $\pm$ 0.75 | 72.67 $\pm$ 0.31 | 83.27 $\pm$ 0.35 | 85.97 $\pm$ 0.21 |
| Infer. Time | 158.1ms | 416.5ms | 11.9s | 129.6s |
| Storage Space | 129.4MB | 165.7MB | 1.9GB | 1.6GB |
| **NID**$_{SAGE}$ | 76.78 $\pm$ 0.59 | 70.52 $\pm$ 0.14 | 81.83 $\pm$ 0.26 | 85.63 $\pm$ 0.31 |
| Infer. Time | 0.4ms | 0.3ms | 0.7ms | 27.1ms |
| Storage Space | 0.4MB | 1.2MB | 17.5MB | 16.4MB |

Table 3: Graph-level performance in supervised representation learning from LRGB.

| Inductive | Peptides-func | Peptides-struct |
|---|---|---|
| Avg. # nodes | 150.9 | 150.9 |
| Avg. # edges | 307.3 | 307.3 |
| Metric | AP↑ | MAE↓ |
| GT | 0.6326 $\pm$ 0.0126 | 0.2529 $\pm$ 0.0016 |
| GraphGPS | 0.6535 $\pm$ 0.0041 | 0.2500 $\pm$ 0.0012 |
| GRIT | 0.6988 $\pm$ 0.0082 | 0.2460 $\pm$ 0.0012 |
| Exphormer | 0.6527 $\pm$ 0.0043 | 0.2481 $\pm$ 0.0007 |
| Graph ViT | 0.6970 $\pm$ 0.0080 | 0.2449 $\pm$ 0.0016 |
| GCN | 0.6762 $\pm$ 0.0053 | 0.2512 $\pm$ 0.0007 |
| Infer. Time | 471.1ms | 424.9ms |
| **NID**$_{GCN}$ | 0.6608 $\pm$ 0.0058 | 0.2589 $\pm$ 0.0014 |
| Infer. Time | 0.4ms | 0.4ms |

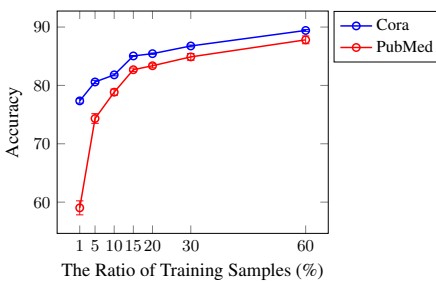

Figure 4: Supervised node classification results of **NID**$_{GCN}$ with varying ratios of training samples.

Table 4: Link prediction in supervised representation learning. The baselines are from Wang et al. (2023c) (%).

| Inductive | Cora | CiteSeer | PubMed | ogbl-collab |
|---|---|---|---|---|
| # nodes | 2,708 | 3,327 | 19,717 | 235,868 |
| # edges | 5,278 | 4,552 | 44,324 | 1,285,465 |
| Metric | HR@100↑ | HR@100↑ | HR@100↑ | HR@50↑ |
| SEAL | 81.71 $\pm$ 1.30 | 83.89 $\pm$ 2.15 | 75.54 $\pm$ 1.32 | 64.74 $\pm$ 0.43 |
| NBFnet | 71.65 $\pm$ 2.27 | 74.07 $\pm$ 1.75 | 58.73 $\pm$ 1.99 | OOM |
| Neo-GNN | 80.42 $\pm$ 1.31 | 84.67 $\pm$ 2.16 | 73.93 $\pm$ 1.19 | 57.52 $\pm$ 0.37 |
| BUDDY | 88.00 $\pm$ 0.44 | 92.93 $\pm$ 0.27 | 74.10 $\pm$ 0.78 | 65.94 $\pm$ 0.58 |
| NCN | 89.05 $\pm$ 0.96 | 91.56 $\pm$ 1.43 | 79.05 $\pm$ 1.16 | 64.76 $\pm$ 0.87 |
| GCN | 85.73 $\pm$ 1.13 | 89.67 $\pm$ 0.81 | 80.36 $\pm$ 0.58 | 64.05 $\pm$ 0.63 |
| Inference Time | 44.2ms | 51.5ms | 102.7ms | 517.1ms |
| **NID**$_{GCN}$ | **90.33** $\pm$ 0.76 | 88.56 $\pm$ 0.72 | 75.67 $\pm$ 0.63 | 64.31 $\pm$ 0.48 |
| Inference Time | 10.9ms | 9.1ms | 22.1ms | 119.7ms |

### 4.1.1 SUPERVISED NODE IDs FOR SUPERVISED REPRESENTATION LEARNING

**Node Classification, Tables 1, 2, Figure 4.** We have conducted extensive evaluations on 8 homophilic and 4 heterophilic graphs, testing scalability on 4 large-scale graphs, each with millions of nodes. Recently, Luo et al. (2024b; 2025b) observed that classic GNNs can achieve competitive performance in node classification with proper hyperparameter tuning. Building on this, we implement our NID on GCN, GAT and SAGE, maintaining all experimental settings as described by Luo et al. (2024b), as they match the hyperparameter search space used in SOTA Polynormer (Deng et al., 2024). We compare NID against SOTA GNNs and Graph Transformers (GTs). Additionally, we compare NID against SOTA GNN-to-MLP methods VQGraph and Graph-MLP under the same settings.

As demonstrated in Table 1, our NID performs competitively with SOTA methods on both homophilic and heterophilic graphs, suggesting that our compact discrete node IDs retain nearly all essential information compared to original GNN node embeddings. Furthermore, the performance comparison clearly shows the superior quality of our NID over SOTA GNN-to-MLP method VQGraph. Notably, NID surpasses all baselines across 4 heterophilic graphs. In a detailed analysis of Questions, we observe that **NID**$_{GCN}$ outperforms GCN by 20%. This dataset represents a highly imbalanced binary classification task, with 98% of nodes classified into the same category. This example illustrates that the node IDs may preserve information beyond that of original GNN node embeddings.

As shown in Table 2, **NID**$_{SAGE}$ not only achieves near-SOTA performance on datasets with millions of nodes but also maintains fast inference times; notably, **NID**$_{SAGE}$ achieves a speed increase ranging from 400× to 17,000× faster than SAGE across 4 large graphs. Remarkably, our IDs require only a small fraction of labels for training. For instance, in the case of the ogbn-products dataset, only 8% of the data is used for training. We analyze the training ratio in Figure 4, showing that merely 15% of the training dataset is sufficient to train node IDs that achieve effective predictive performance.

**Graph Classification, Table 3.** We compare NID against SOTA GNNs and GTs designed for graph-level tasks on two peptide graph benchmarks from LRGB (Dwivedi et al., 2022): Peptides-func and Peptides-struct. We take all evaluation protocols suggested by Rampášek et al. (2022). As evidenced in Table 3, applying pooling to the node IDs achieves excellent performance with notable efficiency improvement, highlighting the potential of NID for supervised learning in graph-level tasks.

Table 5: Attributed graph clustering results; normalized mutual information, and F1-score (%).

| | Cora | | CiteSeer | | PubMed | | Computer | | Photo | | Physics | | ogbn-arxiv | |
| | NMI↑ | F1↑ | NMI↑ | F1↑ | NMI↑ | F1↑ | NMI↑ | F1↑ | NMI↑ | F1↑ | NMI↑ | F1↑ | NMI↑ | F1↑ |
|---|---|---|---|---|---|---|---|---|---|---|---|---|---|---|
| SBM | 36.2 | 30.2 | 15.3 | 19.1 | 16.4 | 16.7 | 48.4 | 34.6 | 59.3 | 47.4 | 45.4 | 30.4 | 31.9 | 28.3 |
| AGC | 34.1 | 28.9 | 25.5 | 27.5 | 18.2 | 18.4 | 51.3 | 35.3 | 59.0 | 44.2 | - | - | - | - |
| SDCN | 27.9 | 29.9 | 31.4 | 41.9 | 19.5 | 29.9 | 24.9 | 45.2 | 41.7 | 45.1 | 50.4 | 39.9 | 15.3 | 28.8 |
| DAEGC | 8.3 | 13.6 | 4.3 | 18.0 | 4.4 | 11.6 | 42.5 | 37.3 | 47.6 | 45.0 | - | - | - | - |
| NOCD | 46.3 | 36.7 | 20.0 | 24.1 | 25.5 | 20.8 | 44.8 | 37.8 | 62.3 | 60.2 | 51.9 | 28.7 | 20.7 | 38.2 |
| DiffPool | 32.9 | 34.4 | 20.0 | 23.5 | 20.2 | 26.3 | 22.1 | 38.3 | 35.9 | 41.8 | - | - | - | - |
| MinCut | 35.8 | 25.0 | 25.9 | 20.1 | 25.4 | 15.8 | - | - | - | - | 48.3 | 24.9 | 36.0 | 27.1 |
| Ortho | 38.4 | 26.6 | 26.1 | 20.5 | 20.3 | 13.9 | - | - | - | - | 44.7 | 23.7 | 35.6 | 26.7 |
| DMoN | 48.8 | 48.8 | 33.7 | 43.2 | 29.8 | 33.9 | 49.3 | 45.4 | 63.3 | 61.0 | 56.7 | 42.4 | 37.6 | 45.7 |
| DGCluster | 62.1 | 54.5 | 41.0 | 32.2 | 32.6 | 34.6 | 60.4 | 52.2 | 77.3 | 75.9 | 65.7 | 49.2 | 31.2 | 32.4 |
| Clustering Time | 93.6ms | | 119.6ms | | 405.5ms | | 286.1ms | | 204.6ms | | 547.4ms | | 2.7s | |
| **NID**$_{\text{DGCluster}}$ | **70.5** | **73.9** | **54.1** | **63.3** | **40.6** | **50.9** | **62.1** | **58.2** | 75.6 | 75.4 | **69.8** | **65.4** | 32.4 | 35.6 |
| Clustering Time | 78.3ms | | 77.2ms | | 292.5ms | | 223.6ms | | 140.6ms | | 442.0ms | | 1.8s | |

Table 6: Node classification results in unsupervised representation learning (%).

| Metric | Cora Accuracy↑ | CiteSeer Accuracy↑ | PubMed Accuracy↑ | dim |
|---|---|---|---|---|
| GAE | 71.5 ±0.4 | 65.8 ±0.4 | 72.1 ±0.5 | 16 |
| DGI | 82.3 ±0.6 | 71.8 ±0.7 | 76.8 ±0.6 | 512 |
| MVGRL | 83.5 ±0.4 | 73.3 ±0.5 | 80.1 ±0.7 | 512 |
| InfoGCL | 83.5 ±0.3 | 73.5 ±0.4 | 79.1 ±0.2 | 512 |
| CCA-SSG | 84.0 ±0.4 | 73.1 ±0.3 | 81.0 ±0.4 | 512 |
| MLP | 57.8 ±0.5 | 54.7 ±0.4 | 73.3 ±0.6 | 500 |
| GraphMAE | 84.2 ±0.4 | 73.4 ±0.4 | 81.1 ±0.4 | 512 |
| **NID**$_{\text{MAE}}$ | 80.8 ±0.7 | **74.2 ±0.6** | 76.4 ±0.8 | **6** |

Table 7: Graph classification results in unsupervised representation learning on TUDataset; Accuracy (%).

| | NCI1 | PROTEINS | DD | MUTAG | COLLAB | RDT-B | RDT-M5K | IMDB-B |
| # graphs | 4,110 | 1,113 | 1,178 | 188 | 5,000 | 2,000 | 4,999 | 1,000 |
| Avg. # nodes | 29.8 | 39.1 | 284.3 | 17.9 | 74.5 | 429.7 | 508.5 | 19.8 |
|---|---|---|---|---|---|---|---|---|
| InfoGraph | 76.2 ±1.0 | 74.4 ±0.3 | 72.8 ±1.7 | 89.0 ±1.1 | 70.6 ±1.1 | 82.5 ±1.4 | 53.4 ±1.0 | 73.0 ±0.8 |
| MVGRL | - | - | - | 89.7 ±1.1 | - | 84.5 ±0.6 | - | 74.2 ±0.7 |
| JOAO | 78.3 ±0.5 | 74.0 ±1.1 | 77.4 ±1.1 | 87.6 ±0.7 | 69.3 ±0.3 | 86.4 ±1.4 | 56.0 ±0.2 | 70.8 ±0.2 |
| GraphMAE | 80.4 ±0.3 | 75.3 ±0.4 | - | 88.1 ±1.3 | 80.3 ±0.5 | 88.0 ±0.2 | - | 75.5 ±0.6 |
| AD-GCL | 69.6 ±0.5 | 73.5 ±0.6 | 74.4 ±0.5 | - | 73.3 ±0.6 | 85.5 ±0.7 | 53.0 ±0.8 | 71.5 ±1.0 |
| GraphCL | 77.8 ±0.4 | 74.3 ±0.4 | 78.6 ±0.4 | 86.8 ±1.3 | 71.3 ±1.1 | 89.5 ±0.8 | 55.9 ±0.2 | 71.1 ±0.4 |
| **NID**$_{\text{CL}}$ | 75.9 ±0.6 | 75.1 ±0.5 | 77.8 ±1.1 | 88.6 ±1.7 | 76.9 ±0.3 | **90.7 ±0.9** | 55.0 ±0.5 | 72.3 ±1.2 |
| AutoGCL | 82.0 ±0.2 | 75.8 ±0.3 | 77.5 ±0.6 | 88.6 ±1.0 | 70.1 ±0.6 | 88.5 ±1.4 | 56.7 ±0.1 | 73.3 ±0.4 |
| **NID**$_{\text{AutoGCL}}$ | 78.2 ±1.5 | **75.9 ±0.6** | 77.2 ±0.9 | **90.4 ±0.8** | 74.5 ±1.1 | 89.8 ±0.7 | 54.2 ±0.6 | 72.4 ±0.8 |

**Link Prediction, Table 4.** We test NID on 4 well-known link prediction benchmarks: Cora, Citeseer, Pubmed and ogbl-collab from the OGB (Hu et al., 2020), following the data splits, evaluation metrics and baselines specified by the NCN (Wang et al., 2023c). The results in Table 4 highlight NID's competitive performance, demonstrating both high accuracy and efficiency in link prediction tasks.

### 4.1.2 Self-supervised Node IDs for Unsupervised Representation Learning

**Attributed Graph Clustering, Table 5.** Attributed graph clustering (Cai et al., 2018) focuses on clustering nodes in an attributed graph, where each node is associated with a set of feature attributes. DGCluster (Bhowmick et al., 2024), utilizes GNNs to optimize modularity for this task, representing the latest SOTA method. In this study, we apply the NID framework within DGCluster, quantizing the embeddings learned by the GCN in DGCluster into node IDs, which are then used for clustering. We select 7 datasets from DGCluster and adopt all experimental settings and baselines as described in DGCluster (Bhowmick et al., 2024). As demonstrated in Table 5, **NID**$_{\text{DGCluster}}$ outperforms all baseline models by a considerable margin on 5 datasets and achieves significantly faster runtime on 7 datasets due to the reduced dimensionality of our node IDs.

**Node Classification, Table 6.** We evaluate the performance of our NID on three standard benchmarks: Cora, CiteSeer, and PubMed (Yang et al., 2016). For this purpose, we employ GraphMAE to provide graph learning objective during the training of node IDs. Specifically, we train a 2-layer GAT following the GraphMAE without supervision, resulting in the generation of 6-dim node IDs, denoted as **NID**$_{\text{MAE}}$. Subsequently, we train an MLP and report the mean accuracy on the test nodes. For the evaluation protocol, we follow all the experimental settings used in GraphMAE (Hou et al., 2022), including data splits and evaluation metrics, using all baselines reported by Hou et al. (2022). Table 6 lists the results. MLP refers to predictions made directly on the initial node features. Notably, **NID**$_{\text{MAE}}$ achieves competitive results in comparison to SOTA self-supervised approaches, and even surpasses all other approaches on CiteSeer. Remarkably, our node IDs are comprised of only 6 discrete codes, with each code having a maximum of 32 possible values. This demonstrates that our NID effectively compresses the node's representation into a concise yet information-rich ID.

**Graph Classification, Tables 7 and 13.** We evaluate our NID framework on 8 datasets from TUDataset (Morris et al., 2020): NCI1, PROTEINS, DD, MUTAG, COLLAB, REDDIT-B, REDDIT-M5K, and IMDB-B, utilizing two different methods, GraphCL (You et al., 2020) and AutoGCL (Yin et al., 2022), as graph learning objectives to guide node IDs pre-training. Specifically, using

Table 8: Comparison of codebook usage rates (%).

Table 9: Average GEDs of 1-hop subgraphs among nodes.

Table 10: Accuracy vs Inference Time.

| Usage rate↑ | Cora | CiteSeer | PubMed |
|---|---|---|---|
| VQGraph | 1.3 | 0.8 | 18.1 |
| **NID**$_{\text{GCN}}$ | 84.7 | 97.9 | 79.1 |
| **NID**$_{\text{GCN}(M=1)}$ | 83.3 | 81.3 | 78.1 |

| GEDs↓ | Cora | CiteSeer | PubMed |
|---|---|---|---|
| Random | 7.21 | 4.83 | 9.61 |
| VQGraph | 6.85 | 4.73 | 9.03 |
| **NID**$_{\text{GCN}}$ | 6.15 | 3.89 | 6.22 |

| Metric | Computer | | ogbn-products | |
|---|---|---|---|---|
| | Acc↑ | Time↓ | Acc↑ | Time↓ |
| GCN | 93.78 | 119.6ms | 82.33 | 12.8s |
| SAGE | 93.59 | 95.7ms | 83.27 | 11.9s |
| VQGraph | 90.28 | 1.4ms | 79.17 | 1.6ms |
| **NID**$_{\text{SAGE}}$ | **93.32** | **0.5ms** | 81.83 | **0.7ms** |

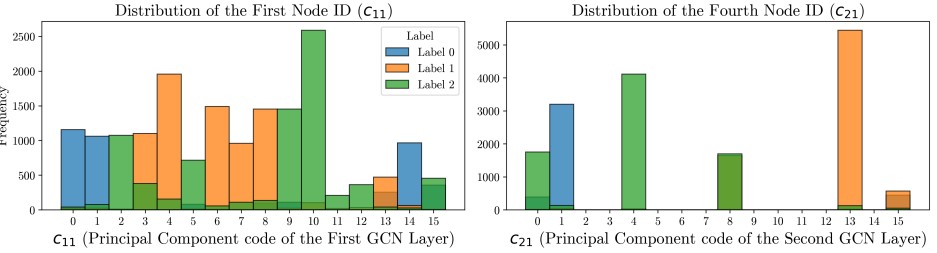

Figure 5: Codeword distributions of $c_{11}$ and $c_{21}$ in PubMed colored by the ground-truth labels.

GraphCL as an example, we employ a GIN with the default settings from GraphCL as the GNN-based encoder, denoted as **NID**$_{\text{CL}}$. We adhere to the evaluation protocol outlined in GraphCL (You et al., 2020). The results are presented in Table 7, where NID outperforms all baselines on 3 out of 8 datasets. This performance demonstrates that our NID is capable of learning meaningful information and demonstrates potential for application in graph-level tasks. Additional linear probing results on MoleculeNet datasets (Wu et al., 2018) are discussed in App. D, showing consistent findings.

## 4.2 ANALYSIS OF NODE IDs

**High Codebook Usage, Table 8.** We calculate the codebook usage rates for VQGraph tokenizer and NID. We find that VQGraph suffers from severe codebook collapse (Dhariwal et al., 2020), where the majority of nodes are quantized into a small number of code vectors, leaving most of the codebook unused. In contrast, our NID achieves high codebook usage, effectively avoiding codebook collapse.

**Qualitative Analysis, Figures 5, 7, 8.** We analyze the supervised node IDs of **NID**$_{\text{GCN}}$ for the PubMed dataset, depicted in Figures 5 & 7. The number of RVQ levels $M$ is set to 3, and the MPNN layers $L$ is set to 2, with a codebook size $K$ of 16. For a given node ID $(c_{11}, c_{12}, c_{13}, c_{21}, c_{22}, c_{23})$ of a node, $0 \leq c_{lm} \leq 15$. The codes $c_{11}$ and $c_{21}$ capture the high-level information of the first and second MPNN layers, respectively. We present the distribution of $c_{11}$ and $c_{21}$ according to different labels. For instance, $c_{11} = 10$ generally corresponds to label "2". Similarly, the majority of nodes with $c_{21} = 13$ are labeled "1". Our node IDs have overlapping codewords for similar labels, allowing the model to effectively share knowledge from similar nodes in the dataset.

**Subgraph Retrieval, Table 9.** We conduct node-centered subgraph retrieval using the supervised node IDs of **NID**$_{\text{GCN}}$ on Cora, CiteSeer and PubMed. We identify the five nodes closest to a query node based on Hamming distances between their node IDs, then compute the average graph edit distance (GED) between the 1-hop subgraph of the query node and the 1-hop subgraphs of these five nodes. The average GED across all nodes is detailed in Table 9. For comparison, we also calculate the GEDs using the VQGraph tokens and the randomly selected nodes. The results show that node IDs perform better in subgraph retrieval, with similar IDs more likely to exhibit similar structures.

**Acceleration in Inference Time, Table 10.** We show the supervised node classification accuracy and model inference time on the Computer and ogbn-products datasets in Table 10. Our results indicate that we achieve the high accuracy of 81% and 93% while maintaining a fast inference time of 0.5ms and 0.7ms, respectively. Since the ogbn-products dataset contains over sixty million edges, graph loading is very slow. However, under the NID framework, we reduce the SAGE inference time from 11.9s to 0.7ms, demonstrating a significant inference speedup of our approach in large networks.

**Ablation Study of the Codebook Size $K$, RVQ Level $M$ and MPNNs Layer $L$, Figure 6.** First, we examine the influence of the codebook size $K$. The optimal $K$ varies across different graphs; however, generally, $K \leq 16$ yields the best performance on most datasets. A larger codebook size may lead to codebook collapse, impairing performance. Second, regarding the RVQ level $M$, we

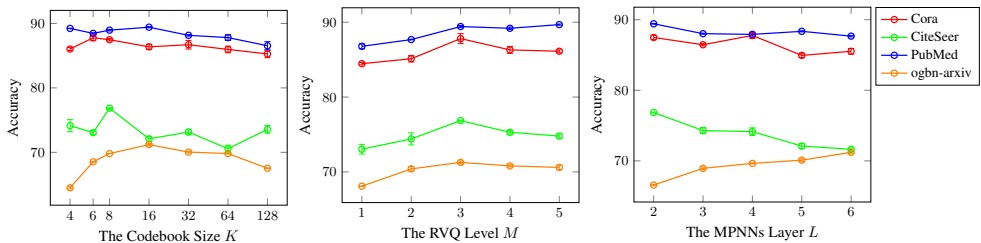

Figure 6: Impact of codebook size, RVQ levels, and MPNN layers on node classification using **NID**GCN.

find that $M = 3$ performs the best, which validates our fixed choice. Notably, when $M = 1$, RVQ degenerates into VQ, which leads to decreased performance. Third, the number of MPNN layers required also differs based on the graph size. For instance, smaller graphs like CiteSeer perform well with just 2 layers, while larger graphs, such as ogbn-arxiv, may require more than 6 layers.

## 5 RELATED WORKS

**Inference Acceleration for GNNs.** GNNs are the preferred method for representation learning on graph-structured data but suffer from decreased inference efficiency as graph size and the number of layers increase, especially in real-time and resource-limited scenarios (Kaler et al., 2022). To address this issue, three main strategies are employed (Ma et al., 2024): knowledge distillation, model pruning, and model quantization. Knowledge distillation includes GNN-to-GNN methods (Yan et al., 2020; Yang et al., 2022) and GNN-to-MLP approaches (Hu et al., 2021; Yang et al., 2024)). Model pruning methods include UGS (Chen et al., 2021) and Snowflake (Wang et al., 2023a)). Lastly, model quantization methods include VQ-GNN (Ding et al., 2021) and QLR (Wang et al., 2023b)).

**Graph Tokenization.** In graph representation learning, significant strides have been made to vectorize structured data for downstream machine learning applications (Chami et al., 2022). Early pioneering efforts, such as DeepWalk (Perozzi et al., 2014) and node2vec (Grover & Leskovec, 2016), popularized the concept of node embedding learning. Subsequently, GNNs have been extensively used as encoders to learn embeddings for graph tokens including nodes, edges, and (sub)graphs, with applications across various domains including molecular motifs (Liu et al., 2024d; Rong et al., 2020; Zhang et al., 2021c; Luo et al., 2023a), recommendation systems (Tang et al., 2024; Liu et al., 2024a), and knowledge graphs (Tang et al., 2023; Lou et al., 2023; Liang et al., 2024b; 2023; 2024a). The emergence of Large Language Models (LLMs) has spurred recent explorations into graph tokenization. Works like InstructGLM (Ye et al., 2023), GraphText (Zhao et al., 2023), and GPT4Graph (Guo et al., 2023) use natural language descriptions of graphs as tokens inputted to LLMs. Additionally, GraphToken (Perozzi et al., 2024) integrates graph tokens generated by GNNs with textual tokens to explicitly represent structured data for LLMs.

**Differences between Our NID and VQGraph.** We clarify that our NID framework is fundamentally different from VQGraph (Yang et al., 2024), even though both approaches utilize VQ techniques to tokenize nodes as discrete codes. VQGraph uses VQ-VAE (Van Den Oord et al., 2017) to obtain soft code assignments, which serve as targets to aid the GNN-to-MLP distillation process. However, similar to other GNN-to-MLP methods, VQGraph lacks the capability to generate interpretable node representations, limiting its application to supervised node classification. In contrast, our NID framework is designed to learn compact discret node representations in both supervised and unsupervised manners, enabling its use across a wide range of downstream tasks. Moreover, VQGraph employs a sizable codebook for tokenization, with a capacity comparable to the size of the input graph, leading to codebook collapse (Tab. 8). In contrast, our NID tokenizer utilizes multiple, small-sized codebooks to achieve a large representational capacity and effectively prevents codebook collapse.

## 6 CONCLUSIONS

We have both empirically and theoretically validated the feasibility of learning highly compact, discrete, and interpretable codes (node IDs) as effective node representations for efficient graph learning. Our proposed NID framework can be seamlessly integrated with state-of-the-art unsupervised and supervised GNN methods to further enhance their performance. These findings have the potential to facilitate graph tokenization and applications involving large language models.

ACKNOWLEDGMENTS

We extend our gratitude to Yiwen Sun for her invaluable assistance. We also express our appreciation to all the anonymous reviewers and ACs for their insightful and constructive feedback. This work received support from National Key R&D Program of China (2021YFB3500700), NSFC Grant 62172026, National Social Science Fund of China 22&ZD153, the Fundamental Research Funds for the Central Universities, State Key Laboratory of Complex & Critical Software Environment (SKLCCSE), and the HK PolyU Grant P0051029. Lei Shi is with School of Computer Science and Engineering, Beihang University, and the State Key Laboratory of Complex & Critical Software Environment.

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

## A    SELF-SUPERVISED NODE IDS USING GRAPHCL

In this section, we discuss GraphCL (You et al., 2020) as a self-supervised learning (SSL) model for self-supervised Node IDs. Contrastive learning aims at learning an embedding space by comparing training samples and encouraging representations from positive pairs of examples to be close in the embedding space while representations from negative pairs are pushed away from each other. Such approaches usually consider each sample as its own class, that is, a positive pair consists of two different views of it; and all other samples in a batch are used as the negative pairs during training.

Specifically, a minibatch of $N$ graphs is randomly sampled and subjected to contrastive learning. This process results in $2N$ augmented graphs, along with a corresponding contrastive loss to be optimized. We redefine $z_{n,i}$ and $z_{n,j}$ for the $n$-th graph in the minibatch. Negative pairs are not explicitly sampled but are instead generated from the other $N-1$ augmented graphs within the same minibatch. The cosine similarity function is denoted as $\text{sim}(z_{n,i}, z_{n,j}) = \frac{z_{n,i}^T z_{n,j}}{\|z_{n,i}\|\|z_{n,j}\|}$. The NT-Xent loss (Sohn, 2016) for the $n$-th graph is then defined as:

$$\mathcal{L}_{\text{GraphCL}} = -\log \frac{\exp(\text{sim}(z_{n,i}, z_{n,j})/\tau)}{\sum_{n'=1, n'\neq n}^{N} \exp(\text{sim}(z_{n,i}, z_{n',j})/\tau)},$$

$$z_{n,i} = \text{R}\left(\text{MPNN}(v_{n,i}, A_{n,i}, X_{n,i})\right),$$

where $\tau$ represents the temperature parameter. The final loss is computed across all positive pairs in the minibatch. Consequently, Equation 7 is reformulated as:

$$\mathcal{L}_{\text{NID}} = \mathcal{L}_{\text{GraphCL}} + \sum_{n\in[1,N]} \sum_{v_{n,i}\in\mathcal{V}_{n,i}} \mathcal{L}_{\text{VQ}}(v_{n,i}) + \sum_{v_{n,j}\in\mathcal{V}_{n,j}} \mathcal{L}_{\text{VQ}}(v_{n,j}), \tag{14}$$

## B    PROOF OF THEOREM 1

*Proof.* For node $v \in \mathcal{V}$ with label $p \in \{0, 1, \cdots, P-1\}$, by data formulation in Section 3.3, we have that there exists at least one node $u \in \mathcal{N}(v)$ that satisfies $x_u = \mu_p$. Given (7), let the first-round clustering center for $v$ be $c_v = c_v^{\parallel} + c_v^{\perp}$, where $c_v^{\parallel}$ is in the direction of $W\mu_p$ and $c_v^{\perp} \perp W\mu_p$. By the assumption on $W$, we know that $W\mu_i \perp W\mu_j$ if $i \neq j$, $i, j \in [P]$, and $W\mu_i = 0$ for $i > P$. Therefore,

$$\|\frac{1}{|\mathcal{N}(v)|} \sum_{u\in\mathcal{N}_v} W x_u - c_v\|^2$$
$$= \|\frac{1}{|\mathcal{N}(v)|} \sum_{u\in\mathcal{N}_v, x_u=\mu_p} W x_u - c_v^{\parallel}\|^2 + \|\frac{1}{|\mathcal{N}(v)|} \sum_{u\in\mathcal{N}_v, x_u\neq\mu_p} W x_u - c_v^{\perp}\|^2. \tag{15}$$

Note that there is no node $u \in \mathcal{N}(v)$ such that $x_u = \mu_j$ for $j \neq p$ and $j \in [P]$. Therefore, we can show that the optimized $c_v^{\parallel}$ by (7) is in the direction of $W\mu_p$, and

$$\|c_v^{\parallel}\| \geq \frac{1}{D_{\mathcal{G}}}, \tag{16}$$

since that

$$\frac{1}{|\mathcal{N}(v)|} \sum_{u\in\mathcal{N}_v} \mathbb{1}[x_u = \mu_p] \geq \frac{1}{D_{\mathcal{G}}}. \tag{17}$$

Hence, we can obtain that the optimized clustering center $c_{v_1}$ and $c_{v_2}$ for $v_1, v_2 \in \mathcal{V}$ with $y_{v_1} \neq y_{v_2}$, we have

$$\|c_{v_1} - c_{v_2}\| \geq \frac{\sqrt{2}}{D_{\mathcal{G}}} \geq \Omega(1), \tag{18}$$

which means that $c_{v_1}$ and $c_{v_1}$ are distinct enough given $D_{\mathcal{G}} \leq O(1)$. Then, we have

$$\text{Node\_ID}(v_1) \neq \text{Node\_ID}(v_2). \tag{19}$$

Then, we analyze the learning process with a linear layer. Let $\boldsymbol{V} = (\boldsymbol{v}_1, \boldsymbol{v}_2, \cdots, \boldsymbol{v}_P)$. The gradient of the loss function against $\boldsymbol{v}_j, j \in [P]$, is computed as

$$
\begin{aligned}
& \frac{1}{|\mathcal{V}_R|} \sum_{v \in \mathcal{V}_R} \frac{\partial \ell(\boldsymbol{z}_v, y_v; \boldsymbol{V})}{\partial \boldsymbol{v}_j} \\
= & \frac{1}{|\mathcal{V}_R|} \sum_{v \in \mathcal{V}_R, y_v = j} -\frac{1}{\hat{p}_{v,j}} \hat{p}_{v,j}(1 - \hat{p}_{v,j}) \boldsymbol{z}_v + \frac{1}{|\mathcal{V}_R|} \sum_{v \in \mathcal{V}_R, y_v = j' \neq j} -\frac{1}{\hat{p}_{v,j}}(-\hat{p}_{v,j}\hat{p}_{v,j'})\boldsymbol{z}_v \\
= & \frac{1}{|\mathcal{V}_R|} \sum_{v \in \mathcal{V}_R, y_v = j} (\hat{p}_{v,j} - 1)\boldsymbol{z}_v + \frac{1}{|\mathcal{V}_R|} \sum_{v \in \mathcal{V}_R, y_v = j' \neq j} \hat{p}_{v,j'}\boldsymbol{z}_v.
\end{aligned}
\tag{20}
$$

Note that $\hat{p}_{v,j} - 1 < 0$, $\hat{p}_{v,j'} > 0$. At iteration 0, $\boldsymbol{V}$ follows a Gaussian distribution and is close to 0, which makes the distribution of $\{\hat{p}_{v,i}\}_{i=1}^P$ close to be uniform. Therefore, after $T$ iterations, we have that

$$
\boldsymbol{v}_j^{(T)} = \boldsymbol{v}_j^{(0)} + \frac{\eta}{|\mathcal{V}_R|} \sum_{v \in \mathcal{V}_R, y_v = j} (1 - \hat{p}_{v,j})\boldsymbol{z}_v - \frac{\eta}{|\mathcal{V}_R|} \sum_{v \in \mathcal{V}_R, y_v = j' \neq j} \hat{p}_{v,j'}\boldsymbol{z}_v,
\tag{21}
$$

where $\eta \leq O(1)$ is the step size. We know that for all the nodes $v \in \mathcal{V}_R$ with $y_v = j$, the first $d_e$ dimensions are the same. Denote the first $d_e$ dimensions of $\boldsymbol{v}_j^{(T)}$ as $\boldsymbol{v}_{j,1:d_e}^{(T)}$. Then, after $T = \Omega(\eta^{-1}P)$ iterations, the magnitude of $\boldsymbol{v}_{j,1:d_e}^{(T)}$ in the direction of the embedding of the first digit of Node_ID$(v)$ is

$$
\Omega(\eta^{-1}P) \cdot \frac{\eta}{\mathcal{V}_R} \sum_{v \in \mathcal{V}_R} \mathbb{1}[y_v = j] \geq \Omega(1).
\tag{22}
$$

Moreover, the magnitude of $\boldsymbol{v}_{j,1:d_e}^{(T)}$ in the direction of the embedding of other digit of Node_ID$(v)$ is

$$
-\Omega(\eta^{-1}P) \cdot \frac{\eta}{\mathcal{V}_R} \sum_{v \in \mathcal{V}_R} \mathbb{1}[y_v = j' \neq j] \cdot \frac{1}{P} \leq -\Omega(\frac{1}{P}).
\tag{23}
$$

Note that the probability of a $d_e$-dimensional embedding for other digits other than the first one is $O(1/K)$. Hence, by $r < K$, we can ensure that the first $d_e$ dimensions are the dominant part for prediction. Therefore, for a given $\boldsymbol{z}_v$ for $v \in \mathcal{G}$, $y_v = j$, we have

$$
\boldsymbol{z}_v^\top \boldsymbol{v}_j^{(T)} - \boldsymbol{z}_v^\top \boldsymbol{v}_{j'}^{(T)} \geq \Omega(1),
\tag{24}
$$

so that

$$
\hat{p}_{v,j} > \hat{p}_{v,j'}.
\tag{25}
$$

We can then derive that for any $v \in \mathcal{V}$,

$$
\mathbb{1}[y_v \neq \arg \max_{i \in [P]} \hat{\boldsymbol{p}}_{v,i}] = 0.
\tag{26}
$$

$\square$

## C DATASETS AND EXPERIMENTAL DETAILS

### C.1 COMPUTING ENVIRONMENT

Our implementation is based on PyG (Fey & Lenssen, 2019) and DGL (Wang et al., 2019b). The experiments are conducted on a single workstation with 8 RTX 3090 GPUs.

### C.2 DESCRIPTION OF DATASETS

Table 11 presents a summary of the statistics and characteristics of the datasets. The initial eight datasets are sourced from TUDataset (Morris et al., 2020), followed by two from LRGB (Dwivedi et al., 2022), and finally the remaining datasets are obtained from Hu et al. (2020); Kipf & Welling (2017); Chien et al. (2020); Pei et al. (2019); Rozemberczki et al. (2021); McAuley et al. (2015); Leskovec & Krevl (2016); Mernyei & Cangea (2020); Lim et al. (2021); Platonov et al. (2023).

Table 11: Overview of the graph learning dataset used in this work (Morris et al., 2020; Dwivedi et al., 2022; Kipf & Welling, 2017; Chien et al., 2020; Pei et al., 2019; Rozemberczki et al., 2021; Hu et al., 2020; McAuley et al., 2015; Leskovec & Krevl, 2016; Mernyei & Cangea, 2020; Lim et al., 2021; Platonov et al., 2023).

| Dataset | # Graphs | Avg. # nodes | Avg. # edges | # Feats | Prediction level | Prediction task | Metric |
|---|---|---|---|---|---|---|---|
| Cora | 1 | 2,708 | 5,278 | 2,708 | node | 7-class classif. | Accuracy |
| Citeseer | 1 | 3,327 | 4,522 | 3,703 | node | 6-class classif. | Accuracy |
| Pubmed | 1 | 19,717 | 44,324 | 500 | node | 3-class classif. | Accuracy |
| Computer | 1 | 13,752 | 245,861 | 767 | node | 10-class classif. | Accuracy |
| Photo | 1 | 7,650 | 119,081 | 745 | node | 8-class classif. | Accuracy |
| CS | 1 | 18,333 | 81,894 | 6,805 | node | 15-class classif. | Accuracy |
| Physics | 1 | 34,493 | 247,962 | 8,415 | node | 5-class classif. | Accuracy |
| WikiCS | 1 | 11,701 | 216,123 | 300 | node | 10-class classif. | Accuracy |
| Squirrel | 1 | 2,223 | 46,998 | 2,089 | node | 5-class classif. | Accuracy |
| Chameleon | 1 | 890 | 8,854 | 2,325 | node | 5-class classif. | Accuracy |
| Amazon-ratings | 1 | 24,492 | 93,050 | 300 | node | 5-class classif. | Accuracy |
| Questions | 1 | 48,921 | 153,540 | 301 | node | 2-class classif. | ROC-AUC |
| ogbn-arxiv | 1 | 169,343 | 1,166,243 | 128 | node | 40-class classif. | Accuracy |
| ogbn-proteins | 1 | 132,534 | 39,561,252 | 8 | node | 112 binary classif. | ROC-AUC |
| ogbn-products | 1 | 2,449,029 | 61,859,140 | 100 | node | 47-class classif. | Accuracy |
| pokec | 1 | 1,632,803 | 30,622,564 | 65 | node | binary classif. | Accuracy |
| ogbl-collab | 1 | 235,868 | 1,285,465 | 128 | edge | link prediction | Hits@50 |
| Peptides-func | 15,535 | 150.9 | 307.3 | 9 | graph | 10-task classif. | AP |
| Peptides-struct | 15,535 | 150.9 | 307.3 | 9 | graph | 11-task regression | MAE |
| NCI1 | 4,110 | 29.87 | 32.30 | 37 | graph | 2-class classif. | Accuracy |
| MUTAG | 188 | 17.93 | 19.79 | 7 | graph | 2-class classif. | Accuracy |
| PROTEINS | 1,113 | 39.06 | 72.82 | 3 | graph | 2-class classif. | Accuracy |
| DD | 1,178 | 284.32 | 715.66 | 89 | graph | 2-class classif. | Accuracy |
| COLLAB | 5,000 | 74.49 | 2457.78 | 1 | graph | 3-class classif. | Accuracy |
| REDDIT-BINARY | 2,000 | 429.63 | 497.75 | 1 | graph | 2-class classif. | Accuracy |
| REDDIT-MULTI-5K | 4,999 | 508.52 | 594.87 | 1 | graph | 5-class classif. | Accuracy |
| IMDB-BINARY | 1,000 | 19.77 | 96.53 | 1 | graph | 2-class classif. | Accuracy |

- Supervised Node Classification: Cora, Citeseer, Pubmed, Computer, Photo, CS, Physics, WikiCS, Amazon-ratings, Questions, Squirrel, Chameleon, ogbn-arxiv, ogbn-proteins, ogbn-products and pokec. For Cora, Citeseer, and Pubmed, we employ a training/validation/testing split ratio of 60%/20%/20% and use accuracy as the evaluation metric, consistent with Pei et al. (2019). For Squirrel, Chameleon, Amazon-ratings and Questions, we adhere to the standard splits and evaluation metrics outlined in Platonov et al. (2023). For the remaining datasets, standard splits and metrics are followed as specified in Luo et al. (2024b). For comprehensive details on these datasets, please refer to the respective studies (Pei et al., 2019; Luo et al., 2024b).

- Supervised Link Prediction: Cora, Citeseer, Pubmed, ogbl-collab. We follow the standard splits and evaluation metrics specified in Wang et al. (2023c), with further details provided therein.

- Supervised Graph Classification: Peptides-func and Peptides-struct. For each dataset, we follow the standard train/validation/test splits and evaluation metrics in Rampášek et al. (2022). For more comprehensive details, readers are encouraged to refer to Rampášek et al. (2022).

- Attributed Graph Clustering: Cora, Citeseer, PubMed, Computer, Photo, Physics, ogbn-arxiv. To evaluate the clustering performance, we adopt two performance measures: NMI and F1, following the approach used in DGCluster (Bhowmick et al., 2024).

- Unsupervised Node Classification: Cora, Citeseer, Pubmed. For each dataset, we follow the standard splits and evaluation metrics in GraphMAE (Hou et al., 2022).

- Unsupervised Graph Classification: NCI1, PROTEINS, DD, MUTAG, COLLAB, REDDIT-B, REDDIT-M5K, and IMDB-B. Each dataset is a collection of graphs where each graph is associated with a label. Each dataset consists of a set of graphs, with each graph associated with a label. For NCI1, PROTEINS, DD, and MUTAG, node labels serve as input features, while for COLLAB, REDDIT-B, REDDIT-M5K, and IMDB-B, node degrees are utilized. In each dataset, we follow exactly the same data splits and evaluation metircs as the standard settings (You et al., 2020).

## C.3 BASELINES

**Attributed Graph Clustering.** We apply baseline methods from DGCluster (Bhowmick et al., 2024): SBM (Peixoto, 2014), AGC Zhang et al. (2019), SDCN (Bo et al., 2020), DAEGC (Wang et al., 2019a), NOCD (Shchur & Günnemann, 2019), DiffPool (Ying et al., 2018), MinCut (Bianchi et al., 2020), Ortho (Bianchi et al., 2020), DMoN (Tsitsulin et al., 2023) and DGCluster (Bhowmick et al., 2024).

**Unsupervised Node Classification.** We utilize all the baselines from GraphMAE (Hou et al., 2022): GAE (Kipf & Welling, 2016), DGI (Veličković et al., 2018b), MVGRL (Hassani & Khasahmadi, 2020), InfoGCL (Xu et al., 2021), CCA-SSG(Zhang et al., 2021a) and GraphMAE (Hou et al., 2022).

**Unsupervised Graph Classification.** We utilize the baselines from GraphCL (You et al., 2020) and AutoGCL (Yin et al., 2022): InfoGraph (Sun et al., 2019), MVGRL (Hassani & Khasahmadi, 2020), GraphCL (You et al., 2020), JOAO (You et al., 2021), GraphMAE (Hou et al., 2022), AD-GCL Suresh et al. (2021) and AutoGCL (Yin et al., 2022).

**Supervised Node Classification.** We compare our method to the following prevalent GNNs and transformer models from Polynormer (Deng et al., 2024): GCN (Kipf & Welling, 2017), SAGE (Hamilton et al., 2017), GAT (Veličković et al., 2018a), APPNP (Gasteiger et al., 2018), GPRGNN (Chien et al., 2020), LINKX (Lim et al., 2021), Polynormer (Deng et al., 2024), SGFormer (Wu et al., 2023). Furthermore, various other GTs like (Kong et al., 2023; Wu et al., 2022; Chen et al., 2022b; Rampášek et al., 2022; Shirzad et al., 2023; Dwivedi et al., 2023; Liu et al., 2023; Zhang et al., 2023a; Kuang et al., 2021; Luo et al., 2024a; Bo et al., 2023; Chen et al., 2022a; Ying et al., 2021; Dwivedi & Bresson, 2020) exist in related surveys (Hoang et al., 2024; Müller et al., 2023), empirically shown to be inferior to the GTs we compared against for node classification tasks.

**Supervised Link Prediction.** We employ all the baselines from NCN (Wang et al., 2023c): GCN (Kipf & Welling, 2017), SEAL (Zhang & Chen, 2018), NBFnet (Zhu et al., 2021), Neo-GNN (Yun et al., 2021), BUDDY (Chamberlain et al., 2022), NCN (Wang et al., 2023c).

**Supervised Graph Classification.** We compare our method to the GCN (Kipf & Welling, 2017). In terms of transformer models, we consider GT(Dwivedi & Bresson, 2020), Graph ViT (He et al., 2023), Exphormer (Shirzad et al., 2023), GraphGPS (Rampášek et al., 2022), and GRIT (Ma et al., 2023).

We report the performance of baseline models using results from their original papers or official leaderboards, where available, as these are derived from well-tuned configurations. For baselines without publicly available results on specific datasets, we tune their hyperparameters within our search space and tune their unique hyperparameters (not present in MPNNs) according to the search space specified in their original papers to attain the best possible results.

## C.4 HYPERPARAMETERS AND REPRODUCIBILITY

Our source code is available at https://github.com/LUOyk1999/NodeID.

**RVQ Implementation Details.** As outlined in Sec. 3.1, RVQ is used to quantize the MPNN multi-layer embeddings of a node. The selection of MPNNs and the number of layers $L$ are tailored to distinct datasets. For the embeddings from each layer, a consistent three-level ($M = 3$) residual quantization is implemented. And cosine similarity serves as the distance metric $|| \cdot ||$ within the RVQ framework. The codebook size $K$ is tuned in $\{4, 6, 8, 16, 32\}$. The $\beta$ is set to 1.

For the hyperparameter selections of our NID framework, in addition to what we have covered, we list other settings in Table 12. The tasks are presented in the following order: attributed graph clustering, unsupervised node classification, unsupervised graph classification, supervised node classification, supervised link prediction, and supervised graph classification. Below we detail the experimental settings for pretraining the node ID.

**Attributed Graph Clustering.** We implement our NID on top of the GCN in DGCluster (Bhowmick et al., 2024). To ensure a fair comparison, we use the same hyperparameters, including the number of layers, learning rate, hidden dimensions, and clustering method, as in DGCluster (Bhowmick et al., 2024).

Table 12: Task-specific hyperparameter settings of NID framework. The tasks are presented in the following order: attributed graph clustering, unsupervised node classification, unsupervised graph classification, supervised node classification, supervised link prediction, and supervised graph classification.

| Dataset | Codebook size $K$ | MPNN | MPNNs layer $L$ | Hidden dim | LR | epoch | MLP layer |
|---|---|---|---|---|---|---|---|
| Cora | 6 | GCN | 3 | 256 | 0.001 | 300 | - |
| Citeseer | 6 | GCN | 3 | 256 | 0.001 | 300 | - |
| Pubmed | 6 | GCN | 3 | 256 | 0.001 | 300 | - |
| Computer | 6 | GCN | 3 | 256 | 0.001 | 300 | - |
| Photo | 6 | GCN | 3 | 256 | 0.001 | 300 | - |
| Physics | 6 | GCN | 3 | 256 | 0.001 | 300 | - |
| ogbn-arxiv | 6 | GCN | 5 | 256 | 0.001 | 300 | - |
| Cora | 32 | GCN | 2 | 1024 | 0.001 | 1500 | 3 |
| Citeseer | 8 | GCN | 2 | 256 | 0.001 | 500 | 3 |
| Pubmed | 16 | GCN | 2 | 128 | 0.0005 | 500 | 3 |
| NCI1 | 4 | GIN | 5 | 32 | 0.01 | 20 | - |
| MUTAG | 16 | GIN | 4 | 32 | 0.01 | 20 | - |
| PROTEINS | 8 | GIN | 3 | 32 | 0.01 | 20 | - |
| DD | 4 | GIN | 4 | 32 | 0.01 | 20 | - |
| COLLAB | 32 | GIN | 5 | 32 | 0.01 | 20 | - |
| REDDIT-BINARY | 4 | GIN | 5 | 32 | 0.01 | 20 | - |
| REDDIT-MULTI-5K | 4 | GIN | 4 | 32 | 0.01 | 20 | - |
| IMDB-BINARY | 8 | GIN | 3 | 32 | 0.01 | 20 | - |
| Cora | 6 | GCN | 4 | 128 | 0.01 | 1000 | 5 |
| Citeseer | 8 | GCN | 2 | 128 | 0.01 | 1000 | 5 |
| Pubmed | 16 | GCN | 2 | 256 | 0.005 | 1000 | 5 |
| Computer | 8 | GAT | 6 | 512 | 0.001 | 1200 | 5 |
| Photo | 4 | GAT | 6 | 512 | 0.001 | 1200 | 4 |
| CS | 16 | GAT | 7 | 512 | 0.001 | 1600 | 4 |
| Physics | 4 | GAT | 5 | 512 | 0.001 | 1600 | 4 |
| WikiCS | 8 | GAT | 8 | 512 | 0.001 | 1000 | 4 |
| Squirrel | 8 | GCN | 6 | 64 | 0.005 | 1000 | 2 |
| Chameleon | 32 | GCN | 3 | 256 | 0.005 | 1000 | 2 |
| Amazon-ratings | 16 | GAT | 12 | 512 | 0.001 | 2500 | 4 |
| Questions | 4 | GAT | 5 | 512 | 0.00003 | 1500 | 4 |
| ogbn-arxiv | 16 | SAGE | 5 | 256 | 0.0005 | 1000 | 4 |
| ogbn-proteins | 4 | SAGE | 4 | 256 | 0.0005 | 1000 | 5 |
| ogbn-products | 16 | SAGE | 5 | 128 | 0.003 | 1000 | 4 |
| pokec | 16 | SAGE | 7 | 256 | 0.0005 | 2000 | 5 |
| Cora | 32 | GCN | 10 | 256 | 0.004 | 150 | 3 |
| Citeseer | 8 | GCN | 10 | 256 | 0.01 | 10 | 3 |
| Pubmed | 8 | GCN | 10 | 256 | 0.01 | 100 | 3 |
| ogbl-collab | 16 | GCN | 5 | 256 | 0.001 | 150 | 3 |
| Peptides-func | 16 | GCN | 6 | 235 | 0.001 | 500 | 5 |
| Peptides-struct | 16 | GCN | 6 | 235 | 0.001 | 250 | 5 |

**Unsupervised Node Classification.** The pretraining hyperparameters are selected within the Graph-MAE's grid search space, as outlined in Table 12. All other experimental parameters, including dropout, batch size, training schemes, and optimizer, etc., align with those used in GraphMAE (Hou et al., 2022).

**Unsupervised Graph Classification.** Similarly, our pretraining hyperparameters in table are determined within GraphCL's grid search space. All other experimental parameters match those used in GraphCL (You et al., 2020). Specially for this task, following GraphCL (You et al., 2020), we input $\mathbf{NID}_{CL}$ codes into a downstream LIBSVM (Chang & Lin, 2011) classifier. And models are trained for 20 epochs and tested every 10 epochs. We conduct a 10-fold cross-validation on every dataset. For each fold, we utilize 90% of the total data as the unlabeled data and the remaining 10% as the labeled testing data. Every experiment is repeated 5 times using different random seeds, with mean and standard deviation of accuracies (%) reported.

**Supervised Node Classification.** The pretraining hyperparameters listed in the table are based on the grid search space from Luo et al. (2024b). All other experimental parameters follow those outlined in the same study.

**Supervised Link Prediction.** Our pretraining hyperparameters in table are chosen from the NCN's grid search space. All other experimental parameters match those used in NCN (Wang et al., 2023c). For the Cora, CiteSeer, and PubMed datasets, we employ the Hadamard product as the readout

function. For ogbl-collab, we use *sum* pooling on the node IDs of the 1-hop common neighbors (Wang et al., 2023c; Barabási & Albert, 1999) to nodes $u$ and $v$ for the edge $(u, v)$.

**Supervised Graph Classification.** All experimental parameters are consistent with those used by Tönshoff et al. (2023); Luo et al. (2025a).

**Applications of Node IDs for Graph Learning**. After obtaining the ID, we train the Multi-Layer Perceptron (MLP) for different tasks, with the number of layers specified in Table 12 and hidden dimensions of either 256 or 512. We utilize the Adam optimizer (Kingma & Ba, 2014) with the default settings. We set a learning rate of either 0.01 or 0.001 and an epoch limit of 1000. The ReLU function serves as the non-linear activation. Further details regarding hyperparameters can be found in the code in the supplementary material. In all experiments, we use the validation set to select the best hyperparameters. All results are derived from 10 independent runs, with mean and standard deviation of results reported.

# D  ADDITIONAL RESULTS

## D.1  QUALITATIVE ANALYSIS

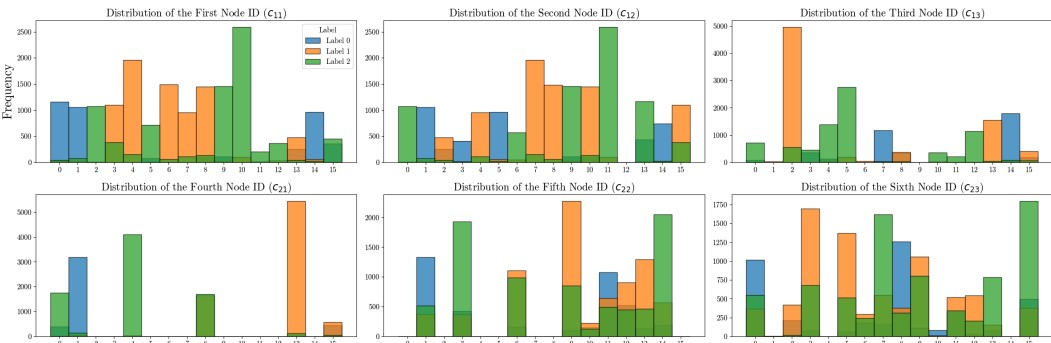

Figure 7: The codeword distributions of the node IDs in PubMed colored by the ground-truth labels.

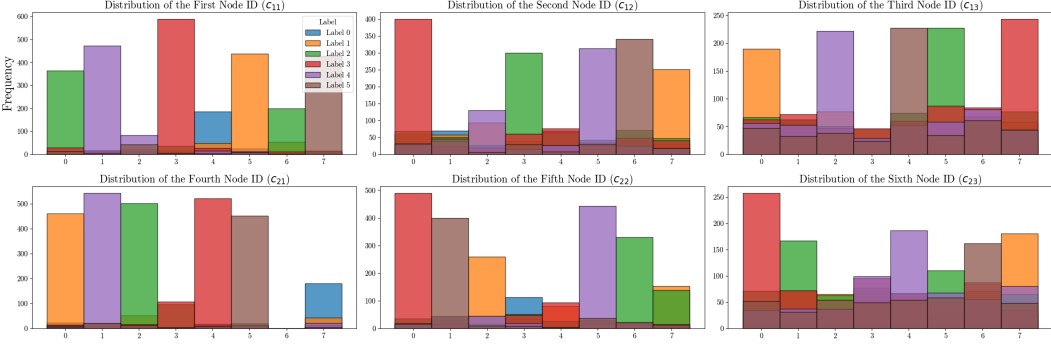

Figure 8: The codeword distributions of the node IDs in CiteSeer colored by the ground-truth labels.

## D.2 ADDITIONAL LINEAR PROBING

**Datasets.** To evaluate the transferability of the proposed method, we test the performance through linear probing on molecular property prediction, adhering to the settings described by You et al. (2020). Initially, the $\text{NID}_{\text{CL}}$ is pre-trained on 2 million unlabeled molecules sourced from ZINC15 (Hu* et al., 2020).

- Node features:
  - Atom number: $[1, 118]$
  - Chirality tag: {unspecified, tetrahedral cw, tetrahedral ccw, other}
- Edge features:
  - Bond type: {single, double, triple, aromatic}
  - Bond direction: {–, endupright, enddownright}

Then, we focus on molecular property prediction, where we adopt the widely-used 7 binary classification datasets contained in MoleculeNet (Wu et al., 2018) for linear probing. The scaffold-split (Ramsundar et al., 2019) is used to split downstream dataset graphs into training/validation/testing set as 80%/10%/10% which mimics real-world use cases. For the evaluation protocol, we run experiments for 10 times and report the mean and standard deviation of ROC-AUC scores (%).

**Model Hyperparameters.** Following You et al. (2020), we adopt a 5-layer GIN (Xu et al., 2018) with a 300 hidden dimension as the MPNN architecture, and set the RVQ codebook size at $K = 16$. We use mean pooling as the readout function. During the pre-training stage, GIN is pre-trained for 100 epochs with batch-size as 256 and the learning rate as 0.001. After the model is trained on the pre-training dataset, it is directly applied to the downstream dataset to obtain node IDs. To evaluate the learned node IDs, we follow the linear probing (linear evaluation) (Akhondzadeh et al., 2023), where a linear classifier (1 linear layer) is trained on the node IDs. During the probing stage, we train for 100 epochs with batch-size as 32, dropout rate as 0.5, and report the test performance using ROC-AUC at the best validation epoch.

The results are presented in Table 13. It is noteworthy that $\text{NID}_{\text{CL}}$ outperforms the baselines in the SIDER, ClinTox, and BBBP datasets, and shows significant improvement over the embeddings from GraphCL. This suggests the robust transferability of $\text{NID}_{\text{CL}}$.

Table 13: Linear probing: molecular property prediction; binary classification, ROC-AUC (%).

|  | Tox21 | ToxCast | Sider | ClinTox | HIV | BBBP | Bace |
|---|---|---|---|---|---|---|---|
| EdgePred | 62.7 ± 0.6 | 55.3 ± 0.4 | 51.0 ± 0.3 | 48.9 ± 6.5 | 64.9 ± 2.0 | 54.8 ± 0.7 | 68.8 ± 0.9 |
| ContextPred | 68.4 ± 0.3 | 59.1 ± 0.2 | 59.4 ± 0.3 | 43.2 ± 1.7 | 68.9 ± 0.4 | 59.1 ± 0.2 | 64.4 ± 0.6 |
| AttrMask | 69.1 ± 0.2 | 58.2 ± 0.2 | 51.7 ± 0.1 | 51.6 ± 0.7 | 60.9 ± 1.3 | 61.0 ± 1.3 | 64.4 ± 2.5 |
| JOAO | 70.6 ± 0.4 | 60.5 ± 0.3 | 57.4 ± 0.6 | 54.1 ± 2.6 | 68.1 ± 0.9 | 63.7 ± 0.3 | 71.2 ± 1.0 |
| SimGRACE | 64.6 ± 0.4 | 59.1 ± 0.2 | 54.9 ± 0.6 | 63.4 ± 2.6 | 66.3 ± 1.5 | 65.4 ± 1.2 | 67.8 ± 1.3 |
| GraphCL | 64.4 ± 0.5 | 59.4 ± 0.2 | 54.6 ± 0.3 | 59.8 ± 1.2 | 63.7 ± 2.3 | 62.4 ± 0.7 | 71.1 ± 0.7 |
| $\text{NID}_{\text{CL}}$ | 66.3 ± 0.4 | 59.1 ± 0.3 | **60.1 ± 0.4** | **65.3 ± 2.2** | 64.3 ± 0.8 | **66.9 ± 0.6** | 66.1 ± 1.2 |

## D.3 ADDITIONAL COMPARISON RESULTS WITH VQGRAPH

**Efficiency.** Our NID employs RVQ at every layer of the MPNN, achieving a representational capacity of $O(K^{M^L})$ with only $O(K \times M \times L)$ in codebook size, which far surpasses VQGraph's capacity that is limited to its single codebook size. As depicted in Table 14, $\text{NID}_{\text{GCN}}$ is considerably more efficient than VQGraph.

**Quality.** We specifically compare the node classification results with VQGraph tokens under the same experimental settings. VQGraph tokens refers to training an MLP with tokens learned by VQGraph tokenizer from Yang et al. (2024). As shown in Table 15, VQGraph tokens lack structural information, likely due to codebook collapse encountered by their tokenizer (see Table 8).

Table 14: Comparison of optimal total codebook sizes.

| | Cora | CiteSeer | PubMed | Computer |
|---|---|---|---|---|
| VQGraph | 2048 | 4096 | 8192 | 16384 |
| $\textbf{NID}_{\text{GCN}}$ ($K \times M \times L$) | $6 \times 3 \times 4 = 72$ | $8 \times 3 \times 2 = 48$ | $16 \times 3 \times 2 = 96$ | $8 \times 3 \times 5 = 120$ |

Table 15: Node classification results in supervised representation learning.

| Metric | Cora Accuracy↑ | CiteSeer Accuracy↑ | PubMed Accuracy↑ | ogbn-arxiv Accuracy↑ |
|---|---|---|---|---|
| VQGraph tokens | 63.39 ± 1.15 | 32.07 ± 2.70 | 54.37 ± 4.76 | 43.57 ± 0.49 |
| $\textbf{NID}_{\text{GCN}}$ | 87.88 ± 0.69 | 76.89 ± 1.09 | 89.42 ± 0.44 | 71.27 ± 0.24 |

## D.4 HIGH CODEBOOK USAGE IN NID

In response to the surprising high codebook usage of our NID compared to VQGraph, we believe this is primarily due to our small codebook size $K$.

As detailed in Table 14, NID achieves high codebook utilization without experiencing codebook collapse. This efficiency is attributed to our smaller codebook size $K$, typically less than 32. In contrast, VQGraph uses a significantly larger codebook size, ranging from 2048 to 32,768 (as reported in Table 12 of the VQGraph paper (Yang et al., 2024)). Their codebook size typically scales the graph size; for example, on the Citeseer dataset, which comprises 2,110 nodes, they used a codebook size of 4,096.

We have observed that a large codebook size often leads to severe codebook collapse. Our use of a small codebook size ensures high codebook usage and prevents codebook collapse. These observations are consistent with findings reported in FSQ (Mentzer et al., 2024).

## D.5 RECONSTRUCTION TASK EXCLUSION IN $\mathcal{L}_{\text{NID}}$

In this section, we address the exclusion of the reconstruction task in our $\mathcal{L}_{\text{NID}}$, differentiating it from other methods such as VQ-VAE and VQGraph. A reconstruction task typically involves using code vectors to regenerate the input data, aiming to minimize the difference between the original input and its reconstruction.

Our decision to omit a reconstruction loss component was based on an ablation study comparing our approach with VQGraph. By replicating VQGraph's experimental setup, minus the reconstruction loss, we found that omitting this component had a negligible impact on performance, as shown in Table 16.

Moreover, as suggested by Vignac et al. (2022), generative models like VQ-VAE, which are primarily designed for continuous data, encounter challenges with graph data due to difficulties in preserving the sparsity and discrete structure inherent in graphs. Therefore, our approach simplifies the model by not involve using the code vectors for a reconstruction task.

Table 16: Impact of excluding reconstruction loss on performance

| Model | Cora | CiteSeer | PubMed | ogbn-products |
|---|---|---|---|---|
| VQGraph | 76.08 ± 0.55 | 78.40 ± 1.71 | 83.93 ± 0.87 | 79.17 ± 0.21 |
| VQGraph without reconstruction loss | 76.17 ± 0.47 | 78.03 ± 1.58 | 83.85 ± 1.24 | 79.23 ± 0.26 |

## D.6 END-TO-END LEARNING WITH RVQ

RVQ can feasibly be applied post-hoc to embeddings generated by a GNN. However, this approach requires a two-phase training process: initially training the GNN and subsequently training the RVQ model. Each phase demands separate optimization of parameters, such as learning rates and epochs.

We conducted experiments to evaluate both methods: end-to-end learning versus post-hoc RVQ application. The results, summarized in Table 17, illustrate that end-to-end learning not only streamlines the training process but also improves performance.

Table 17: Comparison of end-to-end and post-hoc RVQ learning performance

| Method | Cora | Citeseer | PubMed |
|---|---|---|---|
| **NID**$_{\text{GCN}}$ (End-to-End Learning) | $87.88 \pm 0.69$ | $76.89 \pm 1.09$ | $89.42 \pm 0.44$ |
| **NID**$_{\text{GCN}}$ (Post-hoc RVQ) | $86.17 \pm 0.33$ | $75.47 \pm 1.31$ | $88.32 \pm 0.54$ |

### D.7 TRANSDUCTIVE VS. INDUCTIVE INFERENCE

It is important to clarify that our manuscript also incorporates inductive inference evaluations. Notably, the experiments documented in Table 3 & 4 involve inductive graph and edge inference.

For instance, in the Peptides-func dataset detailed in Table 3, we classified 15,535 graphs that were split into training, validation, and test sets. The key point here is that the graphs used during the inference stage were not exposed to the model during training. Additionally, we followed the inductive settings described in the VQGraph paper (Yang et al., 2024) to perform node classification experiments on the ogbn-products dataset. The results in Table 18 clearly demonstrate that our NID method achieves superior performance in both inductive and transductive settings.

Table 18: Comparison of inductive and transductive inference performance on ogbn-products

| Model | ogbn-products (Inductive) | ogbn-products (Transductive) |
|---|---|---|
| VQGraph | $77.50 \pm 0.25$ | $79.17 \pm 0.21$ |
| **NID**$_{\text{SAGE}}$ | $79.13 \pm 0.32$ | $81.83 \pm 0.26$ |

## E ADDITIONAL DISCUSSION

### E.1 RVQ IMPLEMENTATION DETAILS

In our model, each MPNN layer involves the use of $M$ codewords per node, which implies that RVQ is conducted over $M$ iterations.

It is crucial to note that for each MPNN layer, an independent set of $M$ codebooks is utilized. Specifically, we quantize the node embedding $\boldsymbol{h}_v^l$ by sequentially selecting the closest code vector from each codebook:

- The first code vector $\boldsymbol{e}_{c_{l1}}$ ($c_{l1}$ is the codeword) is chosen based on the initial node embedding $\boldsymbol{r}_{l1} = \boldsymbol{h}_v^l$.
- Subsequent code vectors, such as $\boldsymbol{e}_{c_{l2}}$, are selected based on the residual vector $\boldsymbol{r}_{l2} = \boldsymbol{h}_v^l - \boldsymbol{r}_{l1}$, and so forth.

For each codebook approximation, we optimize two types of losses: the codebook loss and the commitment loss, as detailed in Equation 8. Taking the **NID**$_{\text{MAE}}$ as an example, during training, we aggregate all losses from each codebook across all layers ($L \times M$) into an additional loss component. This component is then jointly optimized with the MAE loss, as described in Equation 9, within a single backpropagation step.

### E.2 PRETRAINING RUNTIME

One pertinent aspect of our paper involves analyzing the computational complexity associated with the multi-level optimization of codebooks. To elucidate this, we compared the training time per epoch of our **NID**$_{\text{SAGE}}$ with SAGE on ogbn-products, ogbn-proteins and pokec (see Table 19). The results are revealing: the NID model records a training time of 5.9 seconds per epoch, closely mirroring

that of SAGE, which is at 5.8 seconds per epoch. This comparison substantiates that the additional quantization module integrated into our NID model imposes minimal computational overhead.

These findings underscore the efficiency of our NID model, demonstrating its capability to maintain comparable training times to traditional GNNs despite the added complexity of the multi-level codebook optimization.

Table 19: Empirical training runtime per epoch on ogbn-proteins, ogbn-products, and pokec.

| Dataset | # Nodes | # Edges | Training Time per Epoch |
|---|---|---|---|
| ogbn-proteins | 132,534 | 39,561,252 | SAGE: 0.45s, $\mathbf{NID}_{SAGE}$: 0.51s |
| ogbn-products | 2,449,029 | 61,859,140 | SAGE: 5.8s, $\mathbf{NID}_{SAGE}$: 5.9s |
| pokec | 1,632,803 | 30,622,564 | SAGE: 2.7s, $\mathbf{NID}_{SAGE}$: 3.1s |

### E.3 EXAMPLES OF NODE IDS WITH THE SAME SECOND ID CODE

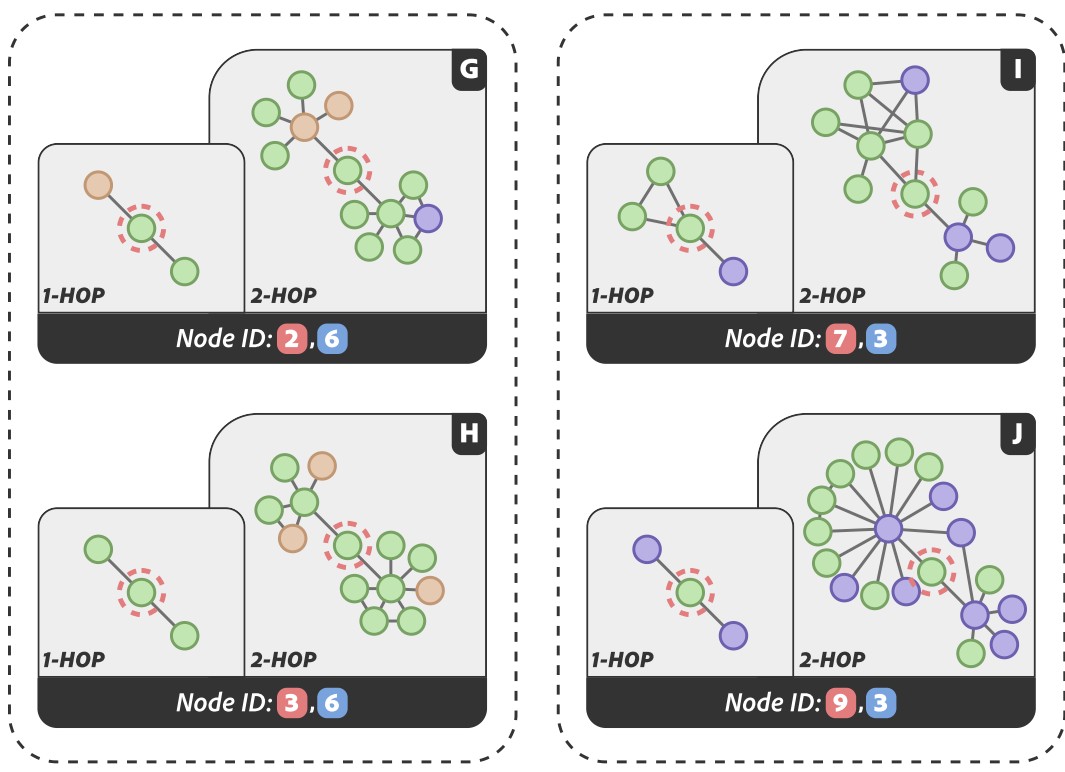

Figure 9: Two real examples of node IDs with the same second ID code from Figure 1 in the PubMed dataset. Nodes G and H have different first ID codes because, despite sharing a similar 1-hop topology, they differ in the label distribution of their 1-hop neighbors. However, they exhibit a similar 2-hop topology and similar label distributions among their 2-hop neighbors, which results in the same second ID code. Additionally, nodes I and J have different first ID codes due to differences in both their 1-hop topologies and the label distributions among their 1-hop neighbors. Nevertheless, their 2-hop neighbors share similar label distributions, leading to the same second ID code, even though their 2-hop topologies are not as similar as those of nodes G and H.

## F LIMITATIONS & BROADER IMPACTS

**Broader Impacts.** This paper presents work whose goal is to advance the field of Machine Learning. There are many potential societal consequences of our work, none which we feel must be specifically highlighted here.

**Limitations.** Our node IDs have proven to be effective in large-scale graphs by accelerating clustering and inference processes due to their low-dimensional nature. However, we have found that the number

of available datasets for very large networks is limited, and we acknowledge that there is still room for extension. Additionally, we believe that node IDs could benefit large language models, a topic we intend to explore more extensively in our future work.

## G    FURTHER RELATED WORKS

**Vector Quantization (VQ).** VQ compresses the representation space into a compact codebook of multiple codewords, using a single code to approximate each vector (Liu et al., 2024b). Advanced methods like VQ-VAE (Van Den Oord et al., 2017) and RQ-VAE (Lee et al., 2022) enhance quantization precision by employing multiple codebooks, initially for image generation and later adapted to recommender systems (Rajput et al., 2024; Liu et al., 2024c) and multimodal representation learning (Zheng et al., 2024; Xia et al., 2024). This paper introduces residual quantization for learning structure-aware node IDs, achieving superior feature compression performance.

**Positional Encodings (PEs) as Graph Tokens.** Transformer models with attention mechanisms can process graphs by tokenizing nodes and edges, incorporating positional or structural graph information through PEs (Müller et al., 2023). The Graph Transformer (Dwivedi & Bresson, 2020) and SAN (Kreuzer et al., 2021) initially employed Laplacian eigenvectors as PEs. Subsequent models like LSPE (Dwivedi et al., 2021) utilized random walk probabilities as node tokens. TokenGT (Kim et al., 2022) introduced orthogonal vectors for both node and edge tokens, and follow-up works also consider larger graphs (Luo et al., 2024c). However, these methods primarily encode the structural information of the graph and overlook the features of nodes, thereby constraining their direct application.

