# OpenReview forum: "Node Identifiers: Compact, Discrete Representations for Efficient Graph Learning"
_ICLR.cc/2025/Conference — ICLR 2025 Poster_

### Official Review · Reviewer_aCEA · 2024-10-29

**Soundness:** 3
**Presentation:** 3
**Contribution:** 3
**Rating:** 6
**Confidence:** 4

**Summary:**

The paper proposes an end-to-end framework NID, which represents nodes with highly compact, discrete vectors, to deal with the inference challenges on large-scale graphs. Specifically, vector quantization is adopted to resident embeddings in each layer of the GNN, and NID can be optimized in either self-supervised or supervised learning ways. Extensive experiments show the effectiveness of the proposed framework.

**Strengths:**

1.	The paper is well-written and solid, especially practical in the case of large-scale graphs.
2.	The overview of the proposed framework helps readers to understand the main pipeline of the work.
3.	The experiment part is sufficient to clarify the effectiveness and efficiency of NID.

**Weaknesses:**

1.	The interpretability of the discrete representations of NID is not discussed in detail in the paper. As the authors point out that the most existed real-valued embeddings often lack interpretability, there should be a detailed discussion.
2.	Some concepts in the paper are vague. Refer to questions for more details.
3.	The introduction of codebook seems useless since the codeword is used in neither loss of VQ nor loss of tasks.

**Questions:**

1.	What does it mean by “share similar 1-hop structures” in line 71 in the description of Figure 1? Why node in case A&B, C&D have similar 1-hop structure? What is the definition of so-called similar 1-hop structures? It is vague.
2.	Let’s extend the first question, so if two nodes have exactly the same second ID code in the figure 1 (but the first ID code is different), does that mean they have “similar 2-hop structures”? If yes, how can they have similar 2-hop structures on the basis of totally different 1-hop neighbors? Which neighbors of the 1-hop neighbors should be considered? If no, what does the same second ID code mean in that case?
3.	The definition of the M is vague. What does M mean in the size of codebook L*M?
4.	In subsection 3.2, the paper says that the Node_ID of each node will be sent into the MLP network, but my question is are you sending the id like “2341” (eg. 4-layer encoder) as the node representation into the MLP network? Why don’t you send the corresponding codeword into the MLP network? I know that code vectors are not used for a reconstruction task, so I am here talking about the downstream tasks.
5.	What will happen if two different nodes have the exactly same Node_ID? Since the size of each codebook is certain, let’s take figure 1 (2-dimensional) as an example, the codebook size of each layer is 5, then you actually can have 25 (5^2) different discrete representations. So there must be some nodes sharing the same Node_ID (collision).
6.	For the argument in box 4.1, as I say, why directly use IDs as representations instead of corresponding vectors in the codebook?
7.	What is the interpretation of your discrete representations?

---

> ### Author Response · Authors · 2024-11-21
> **Authors' Rebuttal 1**
>
> Dear Reviewer aCEA,
>
> We sincerely appreciate your thoughtful, positive, and constructive feedback. We apologize for any confusion that may have arisen from ambiguity in our expression. We hope that the clarification provided below will resolve any misunderstandings and enhance your confidence in our work.
>
> **Q3: Defintion of $M$**
>
> > The definition of the M is vague. What does M mean in the size of codebook L*M?
>
> $M$ refers to the number of levels/tiers in Residual Vector Quantization (RVQ). As stated in lines 175-177, we employ RVQ to quantize the node embeddings and produce $M$ tiers of codes for each node $v$, using $M$ distinct codebooks. Also see lines 128-134 in the updated version. The RVQ process for $L \times M$ codebooks is elaborated in Appendix E.1, where $L$ represents the number of MPNN layers. For the node embeddings produced by each MPNN layer, an $M$-level RVQ is performed. In the training objective (Eq.8), the quantization loss is aggregated across all the $L \times M$ codebooks.
>
> **W3, Q4 & Q6: Code Vectors vs. Node IDs**
>
> >The introduction of codebook seems useless since the codeword is used in neither loss of VQ nor loss of tasks.
>
> If by "codeword" you are referring to code vectors from the codebook, they are indeed integral components of the loss functions. As detailed in Equation 8, the differentiable VQ loss, $L_{VQ}$, is designed to optimize the code vectors ($\mathbf{e}\_{lm}$) and the GNN node embeddings ($\mathbf{r}\_{lm}$) simultaneously. When training is completed, the node IDs (indices of corresponding code vectors) are generated by quantizing the GNN node embeddings using the optimized codebook. Without a well-optimized codebook, meaningful node IDs cannot be generated.
>
> If you refer to "codeword" as the indices of the code vectors, we call them node IDs (see lines 159, 160, and 184). Although these node IDs are not part of any loss functions and are not directly learned through optimization, the codebooks (i.e., the code vectors within them) are optimized. As analysed in Section 3.3, with effectively optimized codebooks, desired performance can be achieved by using the node IDs.
>
> >In subsection 3.2, the paper says that the Node_ID of each node will be sent into the MLP network, but my question is are you sending the id like “2341” (eg. 4-layer encoder) as the node representation into the MLP network? Why don’t you send the corresponding codeword into the MLP network? I know that code vectors are not used for a reconstruction task, so I am here talking about the downstream tasks.
>
> > For the argument in box 4.1, as I say, why directly use IDs as representations instead of corresponding vectors in the codebook?
>
> Your understanding is correct: we use the indices, rather than the code vectors, to train MLPs for downstream tasks. However, it is also possible to use the corresponding code vectors for these tasks, as is common in most compression and quantization methods. These code vectors can perform similarly to the original GNN embeddings, as they have the capability to nearly fully reconstruct the GNN embeddings.
>
> The unique contribution and novelty of our work lies in demonstrating that the indices (node IDs), which are short (mostly 6-15 dim) and discrete (int4 type), can perform competitively with, and sometimes even better than, the original GNN embeddings. It is surprising, yet not unreasonable, that these node IDs are not involved in any loss functions and not directly learned through optimization. This is because the code vectors within codebooks are optimized. Additionally, in Section 3.3, we provide a theoretical analysis to demonstrate the validity of this method. By employing a widely adopted graph data formulation, we prove that the optimized codebook can effectively distinguish nodes. As a result, we can achieve the desired classification performance by using the indices of the corresponding code vectors to train a linear layer.
>
> This finding has significant implications. It suggests that node IDs can serve as effective high-level abstractions of graph data, revealing substantial redundancy in GNN embeddings and providing a level of interpretability that GNN embeddings lack (see further analysis below). This insight may inspire future research in graph tokenization and facilitate graph applications with LLMs.

---

> > ### Author Response · Authors · 2024-11-21
> > **Authors' Rebuttal 2**
> >
> > **Q5: Node ID Collisions**
> >
> > > What will happen if two different nodes have the exactly same Node_ID? Since the size of each codebook is certain, let’s take figure 1 (2-dimensional) as an example, the codebook size of each layer is 5, then you actually can have 25 (5^2) different discrete representations. So there must be some nodes sharing the same Node_ID (collision).
> >
> > Thank you for the thoughtful question.
> >
> > First, as stated in lines 1342-1344, the representational capacity of our NID framework is $K^{M \times L}$, where $M$ denotes the RVQ level, and $L$ denotes the number of MPNN layers (resulting in $M \times L$ codebooks), and $K$ is the size of each codebook. Consequently, the representational capacity scales exponentially with respect to $M \times L$. In our experiments, we set $M = 3$ and $L$ typically ranges from $2$ to $5$, which often results in a large space with a low probability of ID collisions. For example, the size of the representational space for the ogbn-arxiv dataset is $16^{3 \times 5} = 1,152,921,504,606,846,976$.
> >
> > Second, node ID collisions are not necessarily detrimental for downstream tasks. Since node IDs are abstractions of multi-level GNN embeddings, they effectively act as categorical labels of varying granularity for the nodes. Therefore, a collision in node IDs suggests that the nodes may belong to the same class, which can actually aid in classification and clustering tasks.
> >
> >
> > Moreover, our ablation study in Figure 6, Section 4.2 indicates that while increasing $K$ or $L$ can enhance representational capacity (potentially reducing ID collisions), it may adversely affect classification performance. This is because having similar node IDs for nodes in the same class is beneficial for classification.

---

> ### Author Response · Authors · 2024-11-21
> **Authors' Rebuttal 3**
>
> **W1, W2, Q1, Q2 & Q7: Interpretability of Node IDs**
>
>
> > The interpretability of the discrete representations of NID is not discussed in detail in the paper.
>
> > What is the interpretation of your discrete representations?
>
> Thank you for highlighting the importance of discussing the interpretability of discrete node IDs.
>
> Please note that we addressed this in **Section 4.2 ("Qualitative Analysis")**, where we demonstrate the interpretability of node IDs using the PubMed dataset. We would like to offer further explanation here.
>
> Node IDs serve as high-level abstractions of multi-level GNN embeddings and can be considered as *categorical labels of varying granularity* for the nodes. For each node in the PubMed dataset, we learn a 6-dimensional ID ($c_{11},c_{12},c_{13},c_{21},c_{22}, c_{23}$), where $c_{11},c_{12},c_{13}$ are derived from the first-layer GNN embeddings, and $c_{21},c_{22}, c_{23}$ are derived from the second-layer GNN embeddings. Figure 5 illustrates the distributions of $c_{11}$ and $c_{21}$, which are the principle component codes generated by the first-tier RVQ.
>
> The distributions of $c_{11}$ and $c_{21}$ indicate that nodes with the same codes generally belong to the same class. For instance, nodes with $c_{11}=10$ are mostly in the third class (label 2), and nodes with $c_{21}=4$ all belong to the third class. This demonstrates that the codes effectively function as categorical labels. Meanwhile, the distribution of $c_{21}$ is more concentrated, with only 5 values, compared to  $c_{11}$, which takes on all 16 values. This is due to the cumulative smoothing effect of deeper GNN layers, leading to $c_{21}$ exhibiting more concentrated clustering patterns. Therefore, $c_{11}$ can be seen as fine-grained categorical labels, while $c_{21}$ can be considered more coarse-grained labels.
>
> >What does it mean by “share similar 1-hop structures” in line 71 in the description of Figure 1? Why node in case A&B, C&D have similar 1-hop structure? What is the definition of so-called similar 1-hop structures? It is vague.
>
>
> Thank you for your thoughtful question, and we apologize for the confusion. To clarify, we have replaced the term "1-hop structure" with **"1-hop neighborhood"** in the caption of Figure 1 to ensure better understanding.
>
> In Figure 1, the first ID code is derived from the first-layer GNN embeddings, and the second ID code is derived from the second-layer GNN embeddings. *Thus, two nodes sharing the same first ID code suggests that they have similar first-layer GNN embeddings*, which are computed by aggregating the features of their 1-hop neighbors.
>
> A&B, C&D, E&F are three examples that illustrate how nodes with the same first ID code may share "similar 1-hop neighborhoods," which refers to nodes that have *comparable 1-hop topology and similar distributions of labels (or features) among their immediate neighbors*.
>
>
> For nodes E and F, the 1-hop neighborhoods are identical. Nodes A and B have 1-hop neighbors that all share the same green label and exhibit a similar star topology. Similarly, nodes C and D have 1-hop neighbors with green and purple labels in comparable proportions, and these neighbors share a similar, though not identical, topology.
>
> However, nodes with the same first ID code do not necessarily belong to the same class, as demonstrated by the case of E&F, which have different second ID codes.
>
> > Let’s extend the first question, so if two nodes have exactly the same second ID code in the figure 1 (but the first ID code is different), does that mean they have “similar 2-hop structures”? If yes, how can they have similar 2-hop structures on the basis of totally different 1-hop neighbors? Which neighbors of the 1-hop neighbors should be considered? If no, what does the same second ID code mean in that case?
>
> Similarly, *two nodes sharing the same second ID code suggests that they have similar second-layer GNN embeddings*, which are calculated by aggregating the features of neighbors within their 2-hop neighborhood.
>
> We provide two examples in Appendix E.3 that illustrate cases where two nodes share the same second ID code but have different first ID codes.
>
> Nodes G and H have different first ID codes because, despite sharing a similar 1-hop topology, they differ in the label distribution of their 1-hop neighbors. However, they exhibit a similar 2-hop topology and similar label distributions among their 2-hop neighbors, which results in the same second ID code.
>
> Nodes I and J have different first ID codes due to differences in both their 1-hop topologies and the label distributions among their 1-hop neighbors. Nevertheless, their 2-hop neighbors share similar label distributions, leading to the same second ID code, even though their 2-hop topologies are not as similar as those of nodes G and H.
>
> This suggests that the second ID code primarily captures the similarity in the broader 2-hop neighborhood, considering both label distribution and graph topology.

---

> > ### Comment · Reviewer_aCEA · 2024-11-25
> >
> > Thanks for your detailed response. Your reply adequately addresses my concerns, and I hope my suggestions help to improve your paper. After all, I will keep my positive rating.

---

> > > ### Author Response · Authors · 2024-11-25
> > >
> > > Dear Reviewer aCEA,
> > >
> > > Thank you for reaffirming your positive rating. We are pleased that our response has adequately addressed your concerns. Your thoughtful feedback and constructive suggestions have significantly contributed to improving our paper.
> > >
> > > We sincerely appreciate your time and effort. Thank you once again.
> > >
> > > Best regards,
> > >
> > > The Authors

---

### Official Review · Reviewer_w4zV · 2024-11-02

**Soundness:** 3
**Presentation:** 3
**Contribution:** 3
**Rating:** 6
**Confidence:** 4

**Summary:**

This paper introduces a new framework that creates compact and interpretable node representations, called node IDs, using vector quantization to convert GNN embeddings into discrete codes. Experiments across 34 datasets show that these node IDs improve speed and memory efficiency while maintaining competitive performance in various graph tasks.

**Strengths:**

1. The paper introduces a new framework for generating highly compact, discrete node representations (node IDs) that effectively addresses the inference challenges encountered in large-scale graph applications.
2. The authors conduct comprehensive experiments across 34 diverse datasets and tasks, demonstrating the effectiveness of their method.
3. The paper is well-structured and easy to read.

**Weaknesses:**

1. The quantization of node embeddings into discrete node IDs may result in the loss of important structural information, as nuanced variations in node characteristics could be oversimplified into a single codeword.
2. The framework necessitates joint training of the GNN and vector quantization components, which raises concerns about increased time complexity. The authors should provide an analysis of this complexity, particularly focusing on the impact of joint training on computational efficiency and scalability with larger datasets.

**Questions:**

How does the joint training of the GNN and vector quantization components affect the overall computational efficiency, especially when scaling to larger datasets?

---

> ### Author Response · Authors · 2024-11-14
>
> Dear Reviewer w4zV,
>
> We sincerely appreciate your valuable feedback and recognition of our contributions. We hope our response below will further enhance your confidence in our work.
>
> **W1: Potential Loss of Information**
>
> > The quantization of node embeddings into discrete node IDs may result in the loss of important structural information, as nuanced variations in node characteristics could be oversimplified into a single codeword.
>
> This is a valid point. Our approach may lead to some information loss, as the quantized node represenations are highly compact and discrete. Our theoretical and empirical analyses demonstrate that these representations can preserve essential information in GNN embeddings while reducing redundancy, potentially offering a high-level abstraction userful for various downstream tasks. As Reviewer K2Gp noted, "as a VQ paper, the goal is not to really 'beat' anything – just do more with less."
>
> Interestingly, this abstraction can **sometimes exceed the performance of the base GNNs**, as demonstrated in Table 1 with heterophilic datasets and in Table 5's attributed graph clustering tasks. This phenomenon is intriguing and warrants further exploration.
>
>
> **W2 & Q1: Computational Complexity of Joint Training**
>
> > The framework necessitates joint training of the GNN and vector quantization components, which raises concerns about increased time complexity. The authors should provide an analysis of this complexity, particularly focusing on the impact of joint training on computational efficiency and scalability with larger datasets.
> >
> > How does the joint training of the GNN and vector quantization components affect the overall computational efficiency, especially when scaling to larger datasets?
>
> We appreciate your suggestion. In fact, we already analyzed the computational complexity of the joint training of the GNN and vector quantization components in **Appendix E.2**. There, we compared the training time per epoch of our $\text{NID}_\text{SAGE}$ with standard SAGE on the largest dataset used in our experiments, ogbn-products, which contains 2,449,029 nodes. The results show that the training time of our NID model (5.9s) is comparable to that of standard SAGE (5.8s), indicating minimal computational overhead from our vector quantization components.
>
> Following your suggestion, we have also performed additional experiments on another two large-scale datasets, ogbn-proteins and pokec, to further evaluate the efficiency of our approach. The results below confirm that our method scales efficiently without significant increases in computational demands.
>
> | The empirical training runtime per epoch | ogbn-proteins | ogbn-products | pokec      |
> | ---------------------------------------- | ------------- | ------------- | ---------- |
> | \# nodes                                 | 132,534       | 2,449,029     | 1,632,803  |
> | \# edges                                 | 39,561,252    | 61,859,140    | 30,622,564 |
> | SAGE                                     | 0.45s         | 5.8s          | 2.7s       |
> | $\text{NID}_\text{SAGE}$                 | 0.51s         | 5.9s          | 3.1s       |

---

> > ### Comment · Reviewer_w4zV · 2024-11-23
> >
> > Thank the authors for the detailed response. My concerns are well addressed, and I will keep my rating as 6.

---

> > > ### Author Response · Authors · 2024-11-23
> > >
> > > Dear Reviewer w4zV,
> > >
> > > Thank you very much for taking the time to review our rebuttal! We highly appreciate your positive and insightful assessment of our work, as well as your reaffirmation of your rating!
> > >
> > > Best regards,
> > >
> > > The Authors

---

### Official Review · Reviewer_k2Gp · 2024-11-03

**Soundness:** 3
**Presentation:** 3
**Contribution:** 3
**Rating:** 6
**Confidence:** 4

**Summary:**

This paper proposes a vector quantization for GNN representations created from different "depths" of the network at each node.  Its similar to other recently proposed works (I'm most reminded of VQGraph), but its not trained with a reconstruction loss.  Extensive experimentation is provided, but it seems like most baselines are just copied from tables in recent work.  In general the quantization seems to work effectively, performing at about the same level as the original GNN.

Note: Interestingly when a baseline is recomputed by the authors (e.g. GCN) it seems to differ from the result reported in the paper the majority of the results come from (Polynormer).  This raises a substantial red flag, as the author's method should only perform as well as the baseline it quantitizes.  If we instead assume the author's methods perform within epsilon of the corresponding baseline in Polynormer, they would not be superior methods.

**Strengths:**

+ very well written paper about a pressing topic (graph quantization)
+ extensive results help provide details about the method
+ I'm confident the method seems to work well (but perhaps doesn't actually win on baselines, see weaknesses)

**Weaknesses:**

- Egregious experimental result reuse raises a number of inconsistencies for the few methods the authors have recomputed.
- Its difficult to place the results in context with the related work (ie the Polynormer paper results).

**Questions:**

Nice paper.  However when I dove into the experimental results of the Polynormer paper it raised a lot of questions.

1. The core issue is that in this paper you show weak methods (e.g. GCN) performing well on datasets.
Lets take Photos for example.  In Polynormer, GCN is the weakest method (92%).  In your paper, GCN is the strongest method (96%).  This could be because you did a better grid search (good!) but if you don't similarly do a great grid search for the other baselines then I don't trust them at all :)

It makes it very hard understand how your method compares at all to the baselines.  (As a VQ paper the goal is not to really "beat" anything -- just do more with less)

---

> ### Author Response · Authors · 2024-11-14
>
> Dear Reviewer k2Gp,
>
> We sincerely appreciate your careful reading and attention to detail. Your positive comments are truly encouraging. We understand your concern regarding the surprisingly high results of vanilla GNNs, and we would like to address this.
>
> **W1, W2 & Q1: Clarification of Experimental Result**
>
> > Egregious experimental result reuse raises a number of inconsistencies for the few methods the authors have recomputed.
> >
> > Its difficult to place the results in context with the related work (ie the Polynormer paper results).
> >
> > The core issue is that in this paper you show weak methods (e.g. GCN) performing well on datasets. Lets take Photos for example. In Polynormer, GCN is the weakest method (92%). In your paper, GCN is the strongest method (96%). This could be because you did a better grid search (good!) but if you don't similarly do a great grid search for the other baselines then I don't trust them at all :)
> >
> > It makes it very hard understand how your method compares at all to the baselines. (As a VQ paper the goal is not to really "beat" anything -- just do more with less)
>
> Firstly, we would like to address the **misconception** that basic GNNs, such as GCN, are inherently weak models. Recent benchmarking research [1] demonstrates that classic GNNs, including GCN, GAT, and SAGE, are actually **strong** models that can compete with or even surpass the latest state-of-the-art graph transformers (e.g., SGFormer or Polynormer) in node classification, when tuned within the same hyperparameter search space. This suggests that the performance of basic GNNs has been significantly underestimated due to *suboptimal hyperparameter tuning*.
>
> As stated in lines 355-357 of our manuscript, we follow the experimental settings outlined in [1], as they match the hyperparameter search space used in the Polynormer paper. This approach ensures consistency and fair comparison, avoiding conducting a more extensive grid search for GNNs than for other models like Polynormer.
>
> To further resolve your concern, we have also retrained Polynormer using the same hyperparameter search space outlined in [1] on our devices. The results align with those reported in [1], as detailed in the table below. This reaffirms our conclusion that NID performs competitively with SOTA methods on both homophilic and heterophilic graphs.
>
> |                         | Cora↑        | CiteSeer↑    | PubMed↑      | Computer↑    | Photo↑       | CS↑          | Physics↑     | WikiCS↑      | Squirrel↑    | Chameleon↑   | Ratings↑     | Questions↑   |
> | :---------------------- | :----------- | :----------- | :----------- | :----------- | :----------- | :----------- | :----------- | :----------- | :----------- | :----------- | :----------- | ------------ |
> | Polynormer              | 88.11 ± 1.08 | 76.77 ± 1.01 | 87.34 ± 0.43 | 93.18 ± 0.18 | 96.11 ± 0.23 | 95.51 ± 0.29 | 97.22 ± 0.06 | 79.53 ± 0.83 | 40.87 ± 1.96 | 41.82 ± 3.45 | 54.46 ± 0.40 | 78.92 ± 0.89 |
> | Polynormer (retrained)  | 88.32 ± 0.96 | 77.02 ± 1.14 | 88.96 ± 0.51 | 93.78 ± 0.10 | 96.57 ± 0.23 | 95.42 ± 0.19 | 97.18 ± 0.11 | 80.26 ± 0.92 | 41.97 ± 2.14 | 41.97 ± 3.18 | 54.96 ± 0.22 | 78.94 ± 0.78 |
> | GCN                     | 88.77 ± 0.61 | 77.53 ± 0.92 | 90.04 ± 0.25 | 93.78 ± 0.31 | 96.14 ± 0.21 | 95.94 ± 0.28 | 97.36 ± 0.07 | 80.91 ± 0.81 | 44.50 ± 1.92 | 46.11 ± 3.16 | 53.57 ± 0.32 | 77.40 ± 1.07 |
> | $\text{NID}_\text{GCN}$ | 87.88 ± 0.69 | 76.89 ± 1.09 | 89.42 ± 0.44 | 93.41 ± 0.08 | 96.17 ± 0.04 | 95.52 ± 0.10 | 97.34 ± 0.04 | 78.55 ± 0.15 | 45.09 ± 1.72 | 46.29 ± 2.92 | 53.55 ± 0.13 | 96.85 ± 0.10 |
> | GAT                     | 88.22 ± 1.24 | 77.08 ± 0.84 | 89.47 ± 0.25 | 93.53 ± 0.18 | 96.27 ± 0.15 | 94.46 ± 0.14 | 97.17 ± 0.09 | 80.98 ± 0.83 | 38.72 ± 1.46 | 43.44 ± 3.00 | 54.88 ± 0.74 | 78.35 ± 1.16 |
> | $\text{NID}_\text{GAT}$ | 87.35 ± 0.57 | 76.13 ± 1.35 | 88.97 ± 0.36 | 93.38 ± 0.16 | 96.47 ± 0.27 | 94.75 ± 0.16 | 97.13 ± 0.08 | 79.56 ± 0.43 | 37.68 ± 2.04 | 42.83 ± 3.42 | 54.92 ± 0.42 | 97.03 ± 0.02 |
>
> [1] Classic GNNs are Strong Baselines: Reassessing GNNs for Node Classification, NeurIPS 2024 Datasets and Benchmarks Track.

---

> ### Author Response · Authors · 2024-11-21
>
> Dear Reviewer k2Gp,
>
> We are pleased to provide an update addressing your concerns. We have retrained all baseline models in supervised node classification over homophilic and heterophilic graphs, including **GPRGNN, APPNP, SGFormer, and Polynormer**, using the **same hyperparameter tuning search space and training environment** as those employed for our NID model. This ensures **fair and equitable comparisons** across all methods. The retrained results are presented in the table below.
>
> Based on the updated results, we believe it is reasonable to conclude that NID performs competitively with SOTA methods on both homophilic and heterophilic graphs.
>
> |                         | Cora↑        | CiteSeer↑    | PubMed↑      | Computer↑    | Photo↑       | CS↑          | Physics↑     | WikiCS↑      | Squirrel↑    | Chameleon↑   | Ratings↑     | Questions↑   |
> | :---------------------- | :----------- | :----------- | :----------- | :----------- | :----------- | :----------- | :----------- | :----------- | :----------- | :----------- | :----------- | ------------ |
> | GPRGNN (retrained)      | 89.21 ± 0.90 | 77.39 ± 1.23 | 89.37 ± 0.36 | 90.43 ± 0.19 | 95.69 ± 0.20 | 94.24 ± 0.15 | 96.18 ± 0.11 | 80.87 ± 0.81 | 39.44 ± 2.49 | 39.62 ± 3.13 | 52.41 ± 0.57 | 58.63 ± 0.85 |
> | APPNP (retrained)       | 89.98 ± 0.50 | 77.45 ± 0.63 | 88.83 ± 0.20 | 90.32 ± 0.28 | 94.40 ± 0.07 | 94.14 ± 0.26 | 96.84 ± 0.07 | 79.47 ± 0.53 | 36.53 ± 1.79 | 41.38 ± 3.38 | 51.22 ± 0.42 | 75.41 ± 1.03 |
> | SGFormer (retrained)    | 87.83 ± 0.92 | 77.24 ± 0.74 | 89.31 ± 0.54 | 92.42 ± 0.66 | 95.58 ± 0.36 | 95.71 ± 0.24 | 96.75 ± 0.26 | 80.05 ± 0.46 | 42.65 ± 2.41 | 45.21 ± 3.72 | 54.14 ± 0.62 | 73.81 ± 0.59 |
> | Polynormer (retrained)  | 88.32 ± 0.96 | 77.02 ± 1.14 | 88.96 ± 0.51 | 93.78 ± 0.10 | 96.57 ± 0.23 | 95.42 ± 0.19 | 97.18 ± 0.11 | 80.26 ± 0.92 | 41.97 ± 2.14 | 41.97 ± 3.18 | 54.96 ± 0.22 | 78.94 ± 0.78 |
> | GCN                     | 88.77 ± 0.61 | 77.53 ± 0.92 | 90.04 ± 0.25 | 93.78 ± 0.31 | 96.14 ± 0.21 | 95.94 ± 0.28 | 97.36 ± 0.07 | 80.91 ± 0.81 | 44.50 ± 1.92 | 46.11 ± 3.16 | 53.57 ± 0.32 | 77.40 ± 1.07 |
> | $\text{NID}_\text{GCN}$ | 87.88 ± 0.69 | 76.89 ± 1.09 | 89.42 ± 0.44 | 93.41 ± 0.08 | 96.17 ± 0.04 | 95.52 ± 0.10 | 97.34 ± 0.04 | 78.55 ± 0.15 | 45.09 ± 1.72 | 46.29 ± 2.92 | 53.55 ± 0.13 | 96.85 ± 0.10 |
> | GAT                     | 88.22 ± 1.24 | 77.08 ± 0.84 | 89.47 ± 0.25 | 93.53 ± 0.18 | 96.27 ± 0.15 | 94.46 ± 0.14 | 97.17 ± 0.09 | 80.98 ± 0.83 | 38.72 ± 1.46 | 43.44 ± 3.00 | 54.88 ± 0.74 | 78.35 ± 1.16 |
> | $\text{NID}_\text{GAT}$ | 87.35 ± 0.57 | 76.13 ± 1.35 | 88.97 ± 0.36 | 93.38 ± 0.16 | 96.47 ± 0.27 | 94.75 ± 0.16 | 97.13 ± 0.08 | 79.56 ± 0.43 | 37.68 ± 2.04 | 42.83 ± 3.42 | 54.92 ± 0.42 | 97.03 ± 0.02 |

---

> ### Comment · Reviewer_k2Gp · 2024-11-27
> **Thanks**
>
> Thanks to the authors for running additional experiments.  It generally resolves my concerns and I have raised my score accordingly.
>
> However, I must respectfully disagree about the goodness of GCN and GAT.  **They are fundamentally weak models**.  The survey you mention modifies the architectures significantly to address these weaknesses (which is good -- but not relevant to your paper since I suspect you did not perform similar modifications.).

---

> > ### Author Response · Authors · 2024-11-27
> >
> > Dear Reviewer k2Gp,
> >
> > Thank you for taking the time to review our rebuttal and for raising your score. We are pleased to hear that our response has addressed your
> > concerns, and we understand your perspective on GCN and GAT models.
> >
> > Your support is invaluable to us. Thank you once again.
> >
> > Best regards,
> >
> > The Authors

---

### Author Response · Authors · 2024-11-22

Dear Reviewers,

We sincerely thank you for your thoughtful and constructive feedback. We would like to inform you that we have provided individual responses to each of your comments and updated our paper accordingly. We hope that our responses have adequately addressed your concerns, and we look forward to your thoughts.

Thank you for your time and consideration.

Sincerely,

The Authors

---

### Meta-Review · Area_Chair_Wyjh · 2024-12-07

**Metareview:**

The paper introduces a vector quantization (VQ) method for GNN representations, particularly leveraging node embeddings from varying depths of the network. The method is distinct from recent works (e.g., VQGraph) in not relying on a reconstruction loss. The authors provide extensive experimental validation, demonstrating competitive performance across multiple datasets.

The strengths include the presentation, the novelty, the importance of the problem (GNN representation efficiency) and good empirical performance. There is no big weakness. The only weakness is a lack of guarantees of the information loss due to the quantization and how it may affect different applications.

Overall, I lean to accept this paper.

**Additional Comments On Reviewer Discussion:**

Reviewer k2Gp mentioned some discrepancies in the baseline performance reported in this paper and existing studies. The authors have addressed them during the discussion. Other concerns are just some explanations of some proposed concepts and complexity analysis, which has been addressed too.

---

### Decision · Program_Chairs · 2025-01-22

Accept (Poster)